# Nuclear compartmentalization of TERT mRNA and TUG1 lncRNA is driven by intron retention

Gabrijela Dumbović [1,2,7 ✉], Ulrich Braunschweig [3], Heera K. Langner[1], Michael Smallegan [2,4], Josep Biayna[5], Evan P. Hass[1], Katarzyna Jastrzebska [1,8], Benjamin Blencowe [3], Thomas R. Cech [1,2,6], Marvin H. Caruthers[1] & John L. Rinn [1,2,6 ✉]

The spatial partitioning of the transcriptome in the cell is an important form of gene-expression regulation. Here, we address how intron retention influences the spatio-temporal dynamics of transcripts from two clinically relevant genes: TERT (Telomerase Reverse Transcriptase) pre-mRNA and TUG1 (Taurine-Upregulated Gene 1) lncRNA. Single molecule RNA FISH reveals that nuclear TERT transcripts uniformly and robustly retain specific introns. Our data suggest that the splicing of TERT retained introns occurs during mitosis. In contrast, TUG1 has a bimodal distribution of fully spliced cytoplasmic and intron-retained nuclear transcripts. We further test the functionality of intron-retention events using RNA-targeting thiomorpholino antisense oligonucleotides to block intron excision. We show that intron retention is the driving force for the nuclear compartmentalization of these RNAs. For both RNAs, altering this splicing-driven subcellular distribution has significant effects on cell viability. Together, these findings show that stable retention of specific introns can orchestrate spatial compartmentalization of these RNAs within the cell. This process reveals that modulating RNA localization via targeted intron retention can be utilized for RNA-based therapies.

[1] Department of Biochemistry, University of Colorado Boulder, Boulder, CO, USA. [2] BioFrontiers Institute, University of Colorado Boulder, Boulder, CO, USA. [3] Donnelly Centre, University of Toronto, Ontario, Canada. [4] Department of Molecular, Cellular and Developmental Biology, University of Colorado Boulder, Boulder, CO, USA. [5] Institute for Research in Biomedicine, Parc Científic de Barcelona, Barcelona, Spain. [6] Howard Hughes Medical Institute, University of Colorado Boulder, Boulder, CO, USA. [7] Present address: Max Planck Institute of Immunobiology and Epigenetics, Freiburg, Germany. [8] Present address: Department of Bioorganic Chemistry, Centre of Molecular and Macromolecular Studies, Polish Academy of Sciences, Lodz, Poland. ✉email: gabrijela.dumbovic@gmail.com; john.rinn@colorado.edu

Dynamic regulation of subcellular RNA localization is critical for biological processes ranging from organismal development to cellular activity[1–6]. While the underlying mechanisms have been found for some transcripts, new aspects of RNA localization regulation continue to arise[7,8]. Studies have found that splicing affects RNA localization[9,10]. Over the last years, intron retention has emerged as a regulator of the subcellular distribution and nuclear retention of many messenger RNAs and non-coding RNAs[4,11–16]. Nuclear retention of nonfunctional, partially spliced RNAs can serve as a cellular defense mechanism against the translation of RNAs with erroneous splicing. Recent studies show new functions of intron retention such as buffering protein quantity and rapid response to external stimuli[4,12,13,17]. Some intron retention events can be explained by slow post-transcriptional splicing kinetics[18–23]. Alternatively, very specific introns can be stably retained, adding an additional regulatory layer to RNA functionality[24,25]. For instance, intron retention in long noncoding RNAs (lncRNAs) can give rise to transcripts with unique functions in terms of sequence variability and subcellular distribution[26]. Similarly, some coding RNAs retain specific introns, which alters the subcellular localization and availability of their transcripts for translation[27,28].

Recent technological advances enable spatially resolving and quantifying both coding and non-coding RNA distribution on a single-cell, single-transcript, and sub-cellular level[15,29–33]. Here we explored single-molecule localization dynamics of two developmentally and clinically relevant RNAs, telomerase reverse transcriptase (TERT) mRNA, and taurine-upregulated gene 1 (TUG1) lncRNA, across many human cell types. TERT encodes the catalytic subunit of the ribonucleoprotein complex telomerase, which elongates and maintains telomeres[34]. Telomerase is reactivated in most tumors from almost all cancer types, and it is needed for the maintenance of telomeres, which is critical for long-term proliferation of cancer cells[35,36]. The TERT gene is silenced in differentiated cells, hence TERT has been considered as a promising therapeutic target in cancer[37,38]. We and others recently showed that the majority of TERT transcripts are nuclear[39,40], suggesting a potential regulatory mechanism acting at the RNA level.

TUG1 lncRNA has a role in many cellular processes and is associated with malignancies, where it has an oncogenic role (inferred as onco-lncRNA)[41–49]. On a subcellular level, molecular activities of TUG1 lncRNA were pinpointed to the nucleus where it was shown to suppress cell-cycle-related gene expression by interacting with target genomic loci. In the cytoplasm, TUG1 RNA might have a protein-coding role[50,51]. Correspondingly, we and others showed that TUG1 RNA is located in both the nucleus and cytoplasm[31,51,52]. Nevertheless, the mechanism by which TERT and TUG1 maintain dual localization in the nucleus and cytoplasm remains unknown, but could potentially provide important insights into their biology and targeting in cancer. Thus, TERT and TUG1 transcripts represent an opportunity to elucidate when and where splicing occurs, which introns are retained, and how this in turn affects the cellular state. Furthermore, these RNAs provide an opportunity to test if unspliced introns are indeed sufficient to cause nuclear retention, instead of simply being associated with nuclear retention.

We address these questions using two orthogonal approaches: (i) mapping the subcellular localization of TERT and TUG1 fully processed transcripts and splicing intermediates using single-molecule RNA FISH (smRNA FISH) and (ii) using modified antisense oligonucleotides (ASOs) (thiomorpholinos, TMOs) to direct specific splicing events. We show that retention of specific introns drives the nuclear retention of both TERT pre-mRNA and TUG1 lncRNA transcripts. In the case of TERT transcripts, intron retention is regulated during the cell cycle, with two specific introns retained during interphase and spliced out during mitosis. TUG1 transcripts have two distinct populations of fully spliced, cytoplasmic, and intron-retained, nuclear RNAs. To test the functional significance of these intron-retention events, we developed a TMO approach that further drives intron retention and results in a shift in the subcellular localization of the targeted RNA. Altering the nuclear-cytoplasmic distribution of TERT and TUG1 transcripts had significant functional consequences on a cellular scale and reduced cell viability in vitro. Collectively, our findings provide new evidence for a causative relationship between intron retention and spatio-temporal regulation of non-coding and protein-coding RNAs. We suggest the modulation of RNA localization by modified ASOs as a potential therapeutic approach.

## Results

**Nuclear TUG1 lncRNA and TERT mRNA retain introns.** We and others previously observed that TERT transcripts are unexpectedly more abundant in the nucleus than the cytoplasm[39,40]. Somewhat similarly, smRNA FISH revealed that the TUG1 lncRNA is evenly distributed between the nucleus and cytoplasm (Fig. 1)[31,51]. We sought to determine molecular features or splicing patterns that could differentiate nuclear versus cytoplasmic localization of these transcripts. By analyzing available poly(A)$^+$ RNA-seq data from the ENCODE consortium[53], high read coverage across both TUG1 introns were observed (Fig. 1a and Supplementary Fig. 1a), while TERT had high read coverage across three of its introns, intron 1, intron 11 and intron 14 (Fig. 1b and Supplementary Fig. 1b). Next, we calculated the splicing efficiency of each intron using published poly(A)$^+$ RNA-seq data from human induced pluripotent stem (iPS) cells. Percent intron retention (PIR) was calculated using vast-tools[54] as described previously[11]. Briefly, intron retention was evaluated as the ratio of read counts mapping to exon–intron junctions relative to the total number of exon–intron junction reads plus spliced exon-exon junction reads (see "Methods"). The results show that TERT specifically retains introns 11 and 14 in iPS cells (PIR of 30% and 31%, respectively) while the other introns are efficiently spliced (Fig. 1c and Supplementary data 1; of note, intron 1 did not have enough mappable reads to estimate PIR with vast-tools). Further, we confirmed the retention of the first intron in TUG1; it has a PIR of 46% in iPS cells whereas TUG1 intron 2 is absent from the VastDB human database.

We hypothesized that the transcripts with retained introns would be nuclear localized. To determine RNA localization, we designed smRNA FISH probes tiling across TUG1 and TERT exons and introns (Fig. 1a, b). Dual-color smRNA FISH probes independently targeted TUG1 exons and intron 1 or intron 2, and TERT exons and intron 11 or intron 14 (Fig. 1d). We further applied smRNA FISH against TERT intron 2 and GAPDH intron 2 as controls for non-retained introns. Imaging datasets were processed, and co-localized exon and intron spots were quantified as intron-retaining transcripts, while the exon-only signal was quantified as transcripts with that specific intron spliced out (Fig. 1d).

RNA imaging on a human iPS cell line showed an even distribution of TUG1 in the nucleus and cytoplasm (average ~48% and ~52%, respectively) (Fig. 1e). The majority of the transcripts in the nuclear fraction had retained introns, whereas those in the cytoplasmic fraction did not (Fig. 1e). More specifically, the average nuclear PIR (expressed as a percentage of nuclear intron-retaining transcripts over total nuclear transcripts) for TUG1 intron 1 and intron 2 in iPS cells was 62% and 56%, respectively, with significant correlations between the magnitude of detected intron retention and nuclear TUG1 transcript levels ($R^2 = 0.63$ intron 1, $R^2 = 0.47$ intron 2;

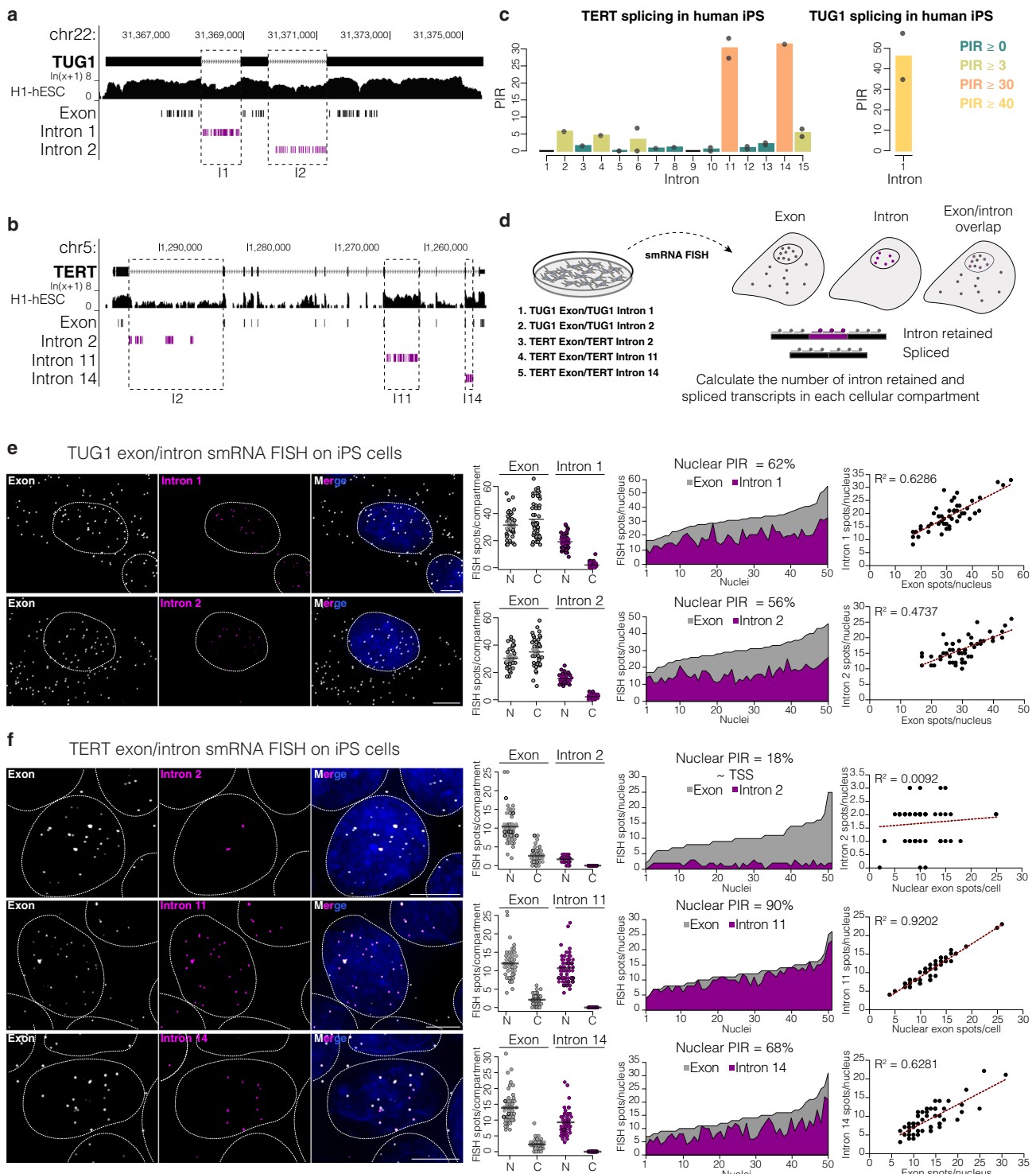

$P = 6.77 \times 10^{-12}$ and $P = 3.33 \times 10^{-8}$, respectively, Pearson correlation, Fig. 1e).

TERT transcripts were retained in the nucleus to an even higher degree than TUG1 transcripts, with on average 86% of the total detected TERT RNAs retained in the nucleus of iPS cells (Fig. 1f). Intron 11 of TERT had a high nuclear PIR (90%) which correlates with the quantity of detected nuclear TERT RNA ($R^2 = 0.92$, $P < 2.2 \times 10^{-16}$, Pearson correlation). Intron 14 was also retained, albeit at a lower proportion (nuclear PIR = 68%), and it also showed a significant correlation with the quantity of nuclear TERT ($R^2 = 0.63$, $P = 7.1 \times 10^{-12}$, Pearson correlation). These

results indicate that TERT intron 11 might have a greater impact on the nuclear retention of TERT RNA than intron 14. TERT intron 2, a control for a non-retained intron in iPS cells, had an average nuclear PIR = 18% with no correlation with the quantity of nuclear TERT ($R^2 = 0.0092$, $P = 0.48$, Pearson correlation). GAPDH intron 2 smRNA FISH showed on average between 1 and 2 punctate signals per cell, which overlapped with GAPDH exon signal and marked the active transcription sites, supporting the specificity of the smRNA FISH approach detecting intron retention in TUG1 and TERT (Supplementary Fig. 1c). Of note, with smRNA FISH we can detect the absolute number of

**Fig. 1 Retention of specific introns correlates with nuclear localization of TERT mRNA and TUG1 lncRNA in hES/iPS cells. a** UCSC Genome Browser showing the TUG1 locus (hg19) and the poly(A)$^+$ RNA-seq track from human ES cells from ENCODE. Below, the location of probes used in smRNA FISH. Exon probes, gray; intron probes, magenta. **b** UCSC Genome Browser showing the TERT locus (hg19) and the poly(A)$^+$ RNA-seq track from human ES cells from ENCODE. Below, the location of probes used in smRNA FISH. Exon probes, gray; intron probes, magenta. **c** Percentage of intron retention of TERT (left) and TUG1 (right) in human iPS cells obtained with vast-tools analysis of RNA-seq data, $n = 2$ poly(A)$^+$ RNA-seq and 1 ribo-depleted RNA-seq. Bars, means across replicates; dots, individual replicates. Introns with insufficient read coverage are shown as black lines in TERT plot. TUG1 intron 2 was absent in the VastDB database. **d** SmRNA FISH scheme. Co-localizing exon and intron signals are considered as unspliced, exon-only signal as spliced. **e** Maximum intensity projections of representative images of TUG1 exon/intron smRNA FISH on iPS cells. Exon, gray; intron 1 and 2, magenta. Nucleus, blue, outlined with a dashed circle. Scale bar, 5 μm. Towards the right: quantification of total and unspliced transcripts for each intron in the nucleus (N) and cytoplasm (C), solid line represents the mean; quantity of spliced (exon only, gray) and intron-retained (magenta) transcripts across 50 nuclei, ordered from fewest to most exon count in the nucleus (average PIR shown on top); correlation between nuclear intron and nuclear TUG1 quantity. $N = 50$ cells, at least two independent RNA FISH stainings. **f** Maximum intensity projections of representative images of TERT exon/intron smRNA FISH on iPS cells. Exon, gray; introns 2, 11, and 14, magenta. Nucleus, blue, outlined with a dashed circle. Scale bar, 5 μm. Towards the right: quantification of total and unspliced transcripts for each intron, solid line represents the mean; quantity of spliced (exon only, gray) and intron-retained (magenta) transcripts across nuclei, ordered from fewest to most exon count in the nucleus (average PIR is shown on top); correlation between nuclear intron and nuclear TERT quantity. $N = 50$ cells (intron 14), 51 cells (intron 2 and 11), at least two independent RNA FISH stainings.

intron-retained and spliced transcripts per cell/compartment in interphase and mitosis. Vast-tools analysis was done on bulk RNA from poly(A)-selected samples, therefore likely under-representing the pool of nuclear unspliced or partially spliced transcripts that we detect by smRNA FISH.

**TUG1 and TERT intron retention across cancer cell types.** The nuclear localization and intron retention observed in healthy iPS cells that endogenously express TUG1 and TERT led us to explore whether this phenomenon also occurs in other cell types and contexts such as cancer, where TERT expression is reactivated and TUG1 is expressed. We performed the same analysis for TUG1 on four cancer cell lines (osteosarcoma U-2 OS, cervical cancer HeLa, colorectal cancer HCT116, and glioblastoma LN-18) and two non-tumor-derived cell types (embryonic kidney HEK293T and BJ fibroblasts); and for TERT on 4 cell lines with TERT re-activation (HeLa, HCT116, HEK293T, and LN-18).

RNA-seq analysis showed high read coverage across TUG1 intron 1 and 2 (Fig. 2a) in all cell lines, indicating that a large fraction of this lncRNA has retained introns. Our smRNA FISH revealed a consistent nuclear/cytoplasmic localization for TUG1 regardless of the cell or cancer type (Fig. 2b and Supplementary Fig. 2a for fibroblasts). As before, there was a significant correlation between the quantity of nuclear TUG1 and intron retention ($R^2 \geq 0.5$, $P < 0.001$ in all cell lines tested, Pearson correlation) (Fig. 2b and Supplementary Fig. 2a for fibroblasts). There were modest differences in the nuclear PIR between cell lines for both intron 1 and intron 2 (mean value ranging from 52 to 75% for intron 1, and from 52 to 67% for intron 2) (Fig. 2c left). Cell lines with higher total PIR of intron 1 and intron 2 tended to have more nuclear TUG1, indicating a correlation between the extent of TUG1 intron retention and nuclear localization ($R^2 = 0.86$ for nuclear enrichment vs. total PIR intron 1; $R^2 = 0.93$ for nuclear enrichment vs. total PIR intron 1, $P = 0.0015$ and $P = 0.0004$, respectively, Pearson correlation) (Fig. 2c right and Supplementary Fig. 2b). Overall, TUG1 showed dual localization and retention of both introns across all analyzed cell lines, with corresponding differences in the ratios of nuclear vs. cytoplasmic TUG1 transcripts and PIR values between cell lines, thereby opening the possibility that this process is being fine-tuned and regulated in a cell-type dependent manner.

We next explored TERT intron retention across cell lines in a similar manner as for TUG1 (Fig. 3). RNA-seq analysis showed high read coverage in introns 11 and 14 (and in HCT116 cells, intron 2), suggesting their retention (Fig. 3a). By smRNA FISH on the HCT116, HEK293T, and LN-18 cell lines we detected TERT expressed in the majority of cells (Fig. 3b). The LN-18 and

HEK293T cell lines showed similar nuclear enrichment of TERT as the iPS cell line (average, 82% and 89% TERT RNA in the nucleus, respectively) (Fig. 3c). This was accompanied by high-retention of intron 11 (nuclear PIR = 91% and 89% for LN-18 and HEK293T, respectively). Retention of intron 11 had a significant correlation with the quantity of nuclear TERT ($R^2 = 0.94$ and 0.96 for LN-18 and HEK293T, respectively, $P < 2.2 \times 10^{-16}$ for both cell lines, Pearson correlation). As in iPS cells, intron 14 was retained to a lesser extent (nuclear PIR = 61% and 47% for LN-18 and HEK293T, respectively) with a modest, but significant, correlation with the quantity of nuclear TERT ($R^2 = 0.28$ and 0.67 for LN-18 and HEK293T, respectively, $P = 7.0 \times 10^{-5}$ and $2.8 \times 10^{-13}$, respectively, Pearson correlation). HeLa cells were excluded from this analysis because they had very few detectable molecules of TERT RNA per cell (Supplementary Fig. 3).

The HCT116 cell line showed some differences compared to iPS, LN-18 and HEK293T cell lines described above. First, very rarely were spliced TERT transcripts detected in the cytoplasm; on average 96% were found in the nucleus (Fig. 3b, c). While intron 2 was not significantly retained in other cell lines examined, HCT116 retained intron 2 in nuclear TERT (nuclear PIR = 78%, $R^2 = 0.91$ with quantity of nuclear TERT, $P < 2.2 \times 10^{-16}$, Pearson correlation), alongside intron 11 (nuclear PIR = 69%, $R^2 = 0.86$ with quantity of nuclear TERT, $P < 2.2 \times 10^{-16}$, Pearson correlation), while intron 14 was retained less efficiently (nuclear PIR = 40%, $R^2 = 0.50$ with quantity of nuclear TERT, $P = 6.5 \times 10^{-9}$, Pearson correlation). This atypical retention of intron 2 can also be observed in the corresponding RNA-seq for HCT116 (Fig. 3a).

Based on our analysis, we find intron 11 robustly retained across different cell lines, while intron 14 showed less and more variable retention, similar to what was observed in iPS cells. Furthermore, these data illustrate the need to analyze possible splicing aberrations that might influence subcellular localization of TERT in certain cell types or cancer, as shown here for intron 2 in the HCT116 cell line.

**Intron retention is conserved across species, but different introns are retained.** We reasoned that if these specific intron retention events for TUG1 and TERT were biologically relevant, they would show evolutionary conservation. To address this, we performed several analyses between human and mouse. The *TUG1* locus has high sequence conservation between human and mouse, maintaining the same gene organization (3 exons and 2 introns) and exhibiting 62.5% overall sequence conservation, 70.2% in exons, and 52.2% in introns (Fig. 4a). We observed that the 5′ exon has slight differences in the annotation compared to human *TUG1* (smaller than the corresponding exon in human)

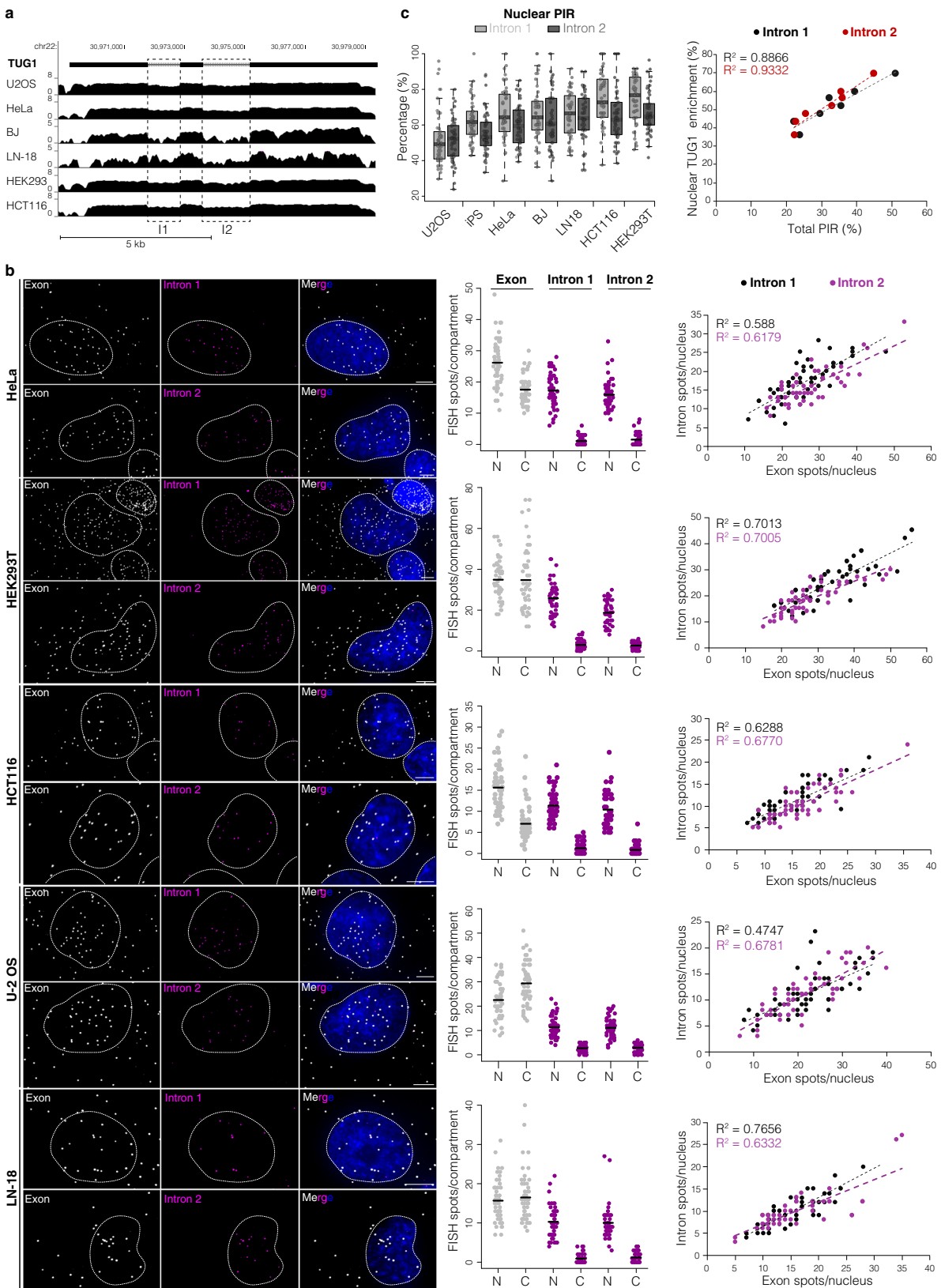

(Fig. 4b). Similarly, the *TERT* locus maintains the same gene organization (16 exons and 15 introns) between human and mouse (Fig. 4a). The overall nucleotide sequence conservation between human and mouse *TERT* loci is only 29.1%, which is mostly due to the low intron sequence conservation (25.5%) while coding sequences exhibit higher conservation (60.7%). Thus, *TUG1* and *TERT* show similar evolutionary conservation in their exonic structures.

We sought to determine whether intron retention is a conserved phenomenon in TUG1 and TERT transcripts in comparable cell

**Fig. 2 TUG1 intron retention is common and fluctuates across cell lines. a** UCSC Genome Browser showing poly(A)$^+$ RNA-seq coverage across TUG1 locus (hg38) from multiple cell lines. Scale ln(x + 1). **b** Maximum intensity projections of representative images of TUG1 exon/intron smRNA FISH across indicated cell lines. Exon, gray; introns 1, 2, magenta; nucleus, blue, outlined with a dashed line. Scale bar, 5 μm. Middle: quantification of total and unspliced transcripts for each intron in the nucleus (N) and cytoplasm (C), solid line represents the mean. Right: correlation between nuclear intron and nuclear TUG1 quantity, intron 1, black; intron 2, magenta. N = 50 cells, at least two independent RNA FISH stainings. **c** Left: nuclear PIR for each intron across cell lines. Midline line, median; lower and upper box limits, 25th and 75th percentiles; whiskers, 1.5 times the interquartile range from the 25th and 75th percentiles. Right: correlation between TUG1 nuclear enrichment and total PIR between different cell lines. Each data point, mean value from one cell line, all measurements shown in Supplementary Fig. 2b. Intron 1, black; intron 2 red. N = 50 cells, at least two independent RNA FISH stainings.

types across species. We analyzed the splicing efficiency of individual Tert and Tug1 introns using published RNA-seq data from mouse iPS (miPS) and mouse embryonic stem (mES) cells (Fig. 4b). In mES and miPS cells, Tug1 intron 1 is not highly retained (PIR of ~6% in miPS and mES), while mouse intron 2 had a higher PIR of ~20%. We applied smRNA FISH to further determine retention of both introns and subcellular localization patterns of Tug1 in mES cells (Fig. 4c). We observed conserved dual localization of Tug1 in the nucleus and cytoplasm (average 61% nuclear Tug1). Intron 2 is highly retained in nuclear Tug1 (PIR = 62%). However, intron 1 is less retained (PIR = 24%), thereby suggesting a more efficient splicing of intron 1 in mouse compared to human TUG1.

Tert introns 11 and 14 are more efficiently spliced in mouse (PIR = 3.4% and 0%, respectively), contrary to their high retention in human cells (PIR = 30.2% and 31.4%, respectively) (Fig. 4b). In contrast, intron 3 and intron 7 are highly retained in mouse Tert (PIR = 24.6% and 23%, respectively, in mES, and 13.3% and 17.2%, respectively, in miPS). Analysis of poly(A) enriched RNA-seq from mES cells showed predominantly nuclear localization of Tert (Fig. 4d).

Intrigued by the retention of different TERT introns between human and mouse, we extended the analysis to seven mammalian species across multiple tissues and cell line samples (Fig. 4e and Supplementary data 2). The analysis revealed an intriguing difference in selective intron retention, even among mammals of the same order. The retention of introns 1 and 11 is conserved across different tissue and cell types in primates. Retention of intron 14 appears to be primarily a human-specific phenomenon. Variability exists in the retention of other introns (for instance, intron 15 is highly retained in macaque). High retention of introns 3 and 7 is conserved between rat and mouse, while rat Tert additionally retains other introns, including intron 11. These data demonstrate that both intron sequences and intron retention have undergone evolutionary divergence between mammalian orders. Intriguingly, TERT contains retained introns in all species investigated, thus opening the possibility that regulation of TERT via subcellular localization may be an evolutionarily conserved mechanism.

We sought to determine more globally whether the observed retention of specific introns in TERT and TUG1 is a common or atypical phenomenon among coding and lncRNA genes. To this end, we analyzed PIR of each intron for every mRNA and lncRNA across hiPS, mES, and miPS cells (Fig. 4f and Supplementary data 3, 4). For each gene, we plotted the maximum PIR among all introns in a given transcript, together with the minimum PIR, if applicable. LncRNAs are known to be spliced less efficiently[11,12,55]. Accordingly, intron retention is generally high in lncRNA genes, extending previous observations that introns in UTRs and non-coding genes are particularly highly retained[11,56]. TUG1 has a maximum PIR typical for a lncRNA gene (Fig. 4f). Interestingly, this analysis revealed that many coding genes have at least one retained intron alongside fully spliced introns. Importantly, PIR of TERT retained introns is among the top 20% of intron retention events in coding genes in both species.

Collectively, these results show that the phenomenon of intron retention of TERT and TUG1 transcripts is conserved across species, but the identity of which introns are retained has evolved.

**Features of retained introns**. We next analyzed intron sequence features that could potentially discriminate retained from efficiently spliced introns. It was shown that retained introns are significantly associated with elevated CG content, reduced length, and relatively weak donor and acceptor splice sites[28]. Introns 3, 7, 11, and 14 are in general longer in human than mouse (Supplementary data 5). No significant differences in GC content were found, except for intron 7 having lower GC content in mouse. We further analyzed the conservation and strength of splice sites of all TERT introns, focusing on highly retained TERT introns in either human or mouse (intron 3, 7, 11, and 14). In all instances, the canonical GT-AG pair is present (Supplementary Fig. 4a). The acceptor and donor splice sites are classified as strong, with no significant differences in the strength of splice sites correlating with intron retention, with the exception of intron 7 which has a weaker donor splice site in mouse (Supplementary Fig. 4b). In contrast, TUG1 introns are highly conserved between human and mouse in length, GC content, and splice site strength (Supplementary Fig. 4, Supplementary data 5).

To begin to assess the influence of RNA binding proteins (RBPs) on intron retention, we analyzed RBP motif instances and binding events detected across all of ENCODE's eCLIP datasets in TERT introns. We reasoned that having highly retained and spliced introns of the same transcript would facilitate discerning binding profiles between spliced and retained introns. The overall binding profile of introns 11 and 14 clustered together in hierarchical clustering of eCLIP peak coverage intron-wide (Supplementary Fig. 5a). This seems to be primarily driven by higher peak coverage indicating increased overall RBP binding in TERT's retained introns (Supplementary Fig. 6), especially around their 5′ and 3′ splice sites, despite their motif density being on par with the other introns (Supplementary Fig. 5b). However, we did not observe proteins that bound uniquely to retained introns 11 and 14, nor proteins that bound all efficiently spliced introns and not those retained. Collectively, these analyses demonstrate that intron 11 and 14 share the property of intron retention and the highest amount of RBP binding. Yet, if the RBP density is due to intron retention or driving intron retention remains unknown.

**Intron-retained nuclear TUG1 and TERT are stable RNAs that remain in the nucleus after transcription termination**. Some RNA processing intermediates retain certain introns due to slow post-transcriptional splicing kinetics[12,23]. To test whether TUG1 and TERT retain introns due to slow splicing kinetics or if those are stable transcripts, we treated cells with Actinomycin D (ActD). ActD inhibits Pol I, Pol II, and Pol III by intercalating in the DNA and preventing transcription elongation[57,58]. Cell lines that endogenously co-express TUG1 and TERT (iPS, LN-18, HEK293T) were treated for up to 4.5 hours and harvested for RT-

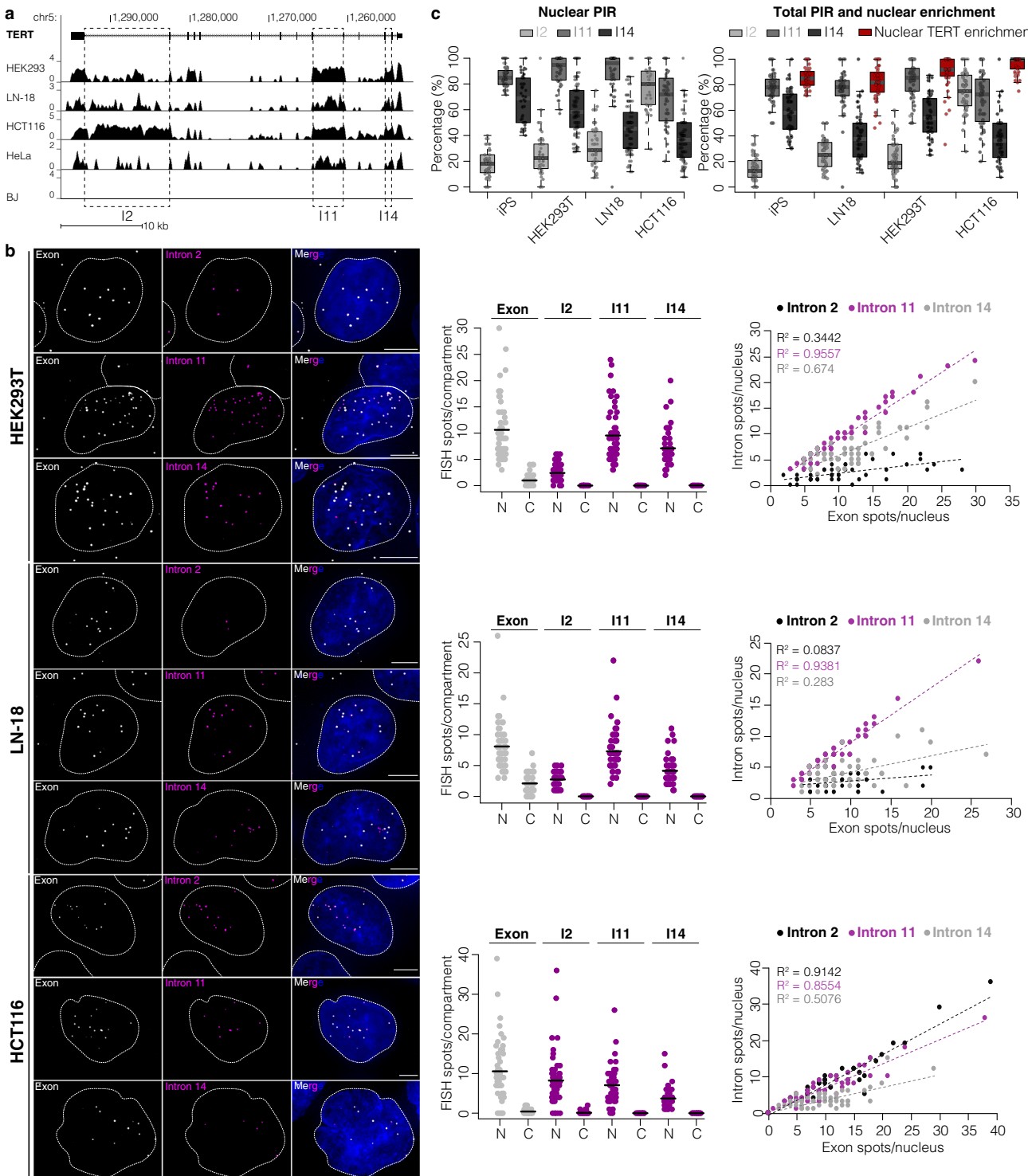

**Fig. 3 Retention of TERT intron 11 is robust across cell lines. a** UCSC Genome Browser showing RNA-seq coverage across TERT locus (hg38) from multiple cell lines. Scale ln(x + 1). **b** Maximum intensity projections of representative images of TERT exon/intron smRNA FISH across different cell lines. Exon, gray; introns 2, 11, 14, magenta; nucleus, blue, outlined with a dashed line. Scale bar, 5 μm. Middle: quantification of total and unspliced transcripts for each intron in the nucleus (N) and cytoplasm (C), solid line represents the mean. Right: correlation between nuclear intron and nuclear TERT count; intron 2, black; intron 11, magenta; intron 14, gray. N = 50 cells, at least two independent RNA FISH stainings. **c** Left: nuclear PIR for each intron across cell lines. Midline line, median; lower and upper box limits, 25th and 75th percentiles; whiskers, 1.5 times interquartile range from the 25th and 75th percentiles. Right: total PIR of each intron and percentage of nuclear enrichment of TERT across cell lines. N = 50 cells, at least two independent RNA FISH stainings.

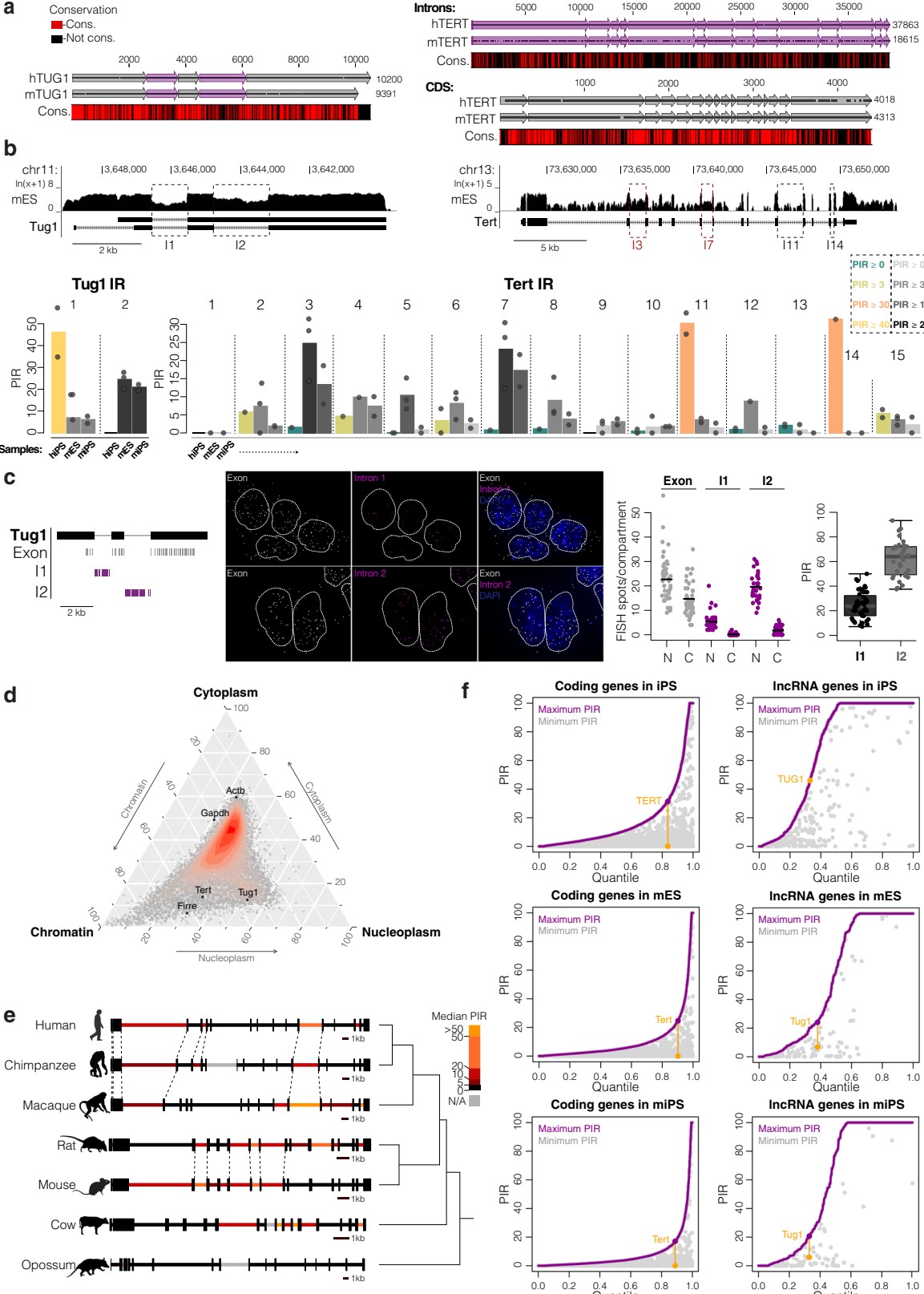

qPCR at several time points (0 h, 40 min, 2.5 h and 4.5 h). We monitored the stability of TUG1 and TERT exons and retained introns. As a stable RNA control, we used GAPDH, while GAPDH intron 2 and pre-ribosomal RNA (45S rRNA) were used as controls for nascent RNAs. We observed an immediate decrease in the nascent 45S rRNA and GAPDH intron 2 after 40 min of ActD treatment. In contrast, in healthy and cancer cell lines, intron-containing TUG1 and TERT RNAs were highly stable even after 4.5 h of transcription inhibition (Fig. 5a). To differentiate nascent transcripts and those that have terminated,

**Fig. 4 Evolutionary conservation of TERT and TUG1 intron retention. a** Alignment of human and mouse *TUG1* locus (left) and *TERT* locus (right). Exons depicted with gray arrows, introns with magenta arrows. Below the alignment is shown a conservation map of conserved (red) and non-conserved (black) nucleotides. **b** UCSC Genome Browser showing poly(A)$^+$ RNA-seq coverage from mouse embryonic stem cells (mES) across the *Tug1* locus and *Tert* locus. Below, percentage intron retention (PIR) of Tert (right) and Tug1 (left) in mouse iPS (miPS) and mES cells obtained with vast-tools analysis on poly(A)$^+$ RNA-seq. Values from human iPS (hiPS) cells are plotted for comparative purposes. Bars indicate means across replicates and dots individual replicates, $n =$ 2 (miPS), 3 (mES). **c** Maximum intensity projections of representative images of Tug1 exon/intron smRNA FISH on mES cells. Exon in gray, introns 1 and 2 in magenta. Nucleus in blue, outlined with a dashed line. Scale bar, 5 μm. Middle: quantification of total and unspliced transcripts for each intron in the nucleus (N) and cytoplasm (C), solid line represents the mean. On the right: percentage of nuclear intron retention (PIR) for intron 1 and intron 2. Midline line, median; lower and upper box limits, 25th and 75th percentiles; whiskers, 1.5 times interquartile range from the 25th and 75th percentiles, $n =$ 44 cells (intron 1), 30 cells (intron 2). **d** Relative subcellular localization of Tert and Tug1 in poly(A)$^+$ RNA-seq from chromatin, cytoplasm and nucleoplasm of mES cells. Cytoplasm-enriched Gapdh and Actb and chromatin-enriched Firre are plotted for comparison. **e** Intron retention of TERT in seven mammalian species. Exon–intron structure is shown and scale bars indicate relative size for each species. Median intron retention across 38 (chimpanzee)—151 (mouse) cell and tissue types is represented by a color scale. Note high retention of introns 2 and 3 in opossum. Dashed lines indicate boundaries of orthologous introns that are retained in both species. Evolutionary relationships are represented by the cladogram on the right. Silhouettes from http://phylopic.org. **f** Cumulative distribution of maximum PIR levels for each coding and lncRNA gene in hiPS, mES and miPS cells (in purple). Minimum PIR value for the same gene is plotted in gray at the same x-axis position. Introns with maximum and minimum PIR values from TERT and TUG1 are connected with a yellow line.

we compared the abundance and stability of intron-retained transcripts in cDNA synthesized with random primers or oligo (dT) during the iPS ActD treatment. The majority of detected TUG1 and TERT intron-retained transcripts had a poly(A) tail in the untreated state and after transcription inhibition, indicating that those transcripts have been 3′-end processed (Fig. 5b).

We applied smRNA FISH during ActD treatment to determine the stability and spatial localization of intron-retained and spliced TUG1 and TERT. Specifically, nuclear TUG1 remained stable across the ActD time course (Fig. 5c for LN-18 and Supplementary Fig. 7a for iPS; no decrease and ~1.2-fold decrease, respectively). In contrast, cytoplasmic TUG1 gradually decreased in both cell lines during ActD time points (~2.2-fold decrease, $P \leq 0.001$, unpaired $t$-test for LN-18; ~2-fold decrease, $P \leq 0.001$, unpaired $t$-test for iPS). Furthermore, retention of intron 1 and 2 remained high even after 4.5 h of treatment, and unspliced TUG1 remained nuclear. Thus, the nuclear, intron-retained TUG1 fraction is more stable than the fully spliced cytoplasmic fraction.

TERT followed a similar trend, with nuclear, unspliced TERT RNA (assessed by retention of intron 11) being highly stable and retained in the nucleus even after 4.5 h of ActD treatment (Fig. 5d and Supplementary Fig. 7b). Nuclear TERT was highly stable during the course of ActD treatment, while cytoplasmic TERT gradually decreased after transcription inhibition (~4-fold decrease, $P \leq 0.001$, unpaired $t$-test for LN-18; ~3-fold decrease, $P \leq 0.001$, unpaired $t$-test for iPS). Retention of intron 11 remained high (no significant decrease) for LN-18 and iPS, and the unspliced transcript remained in the nucleus.

As a control for transcription inhibition, we monitored GAPDH transcription sites visualized by smRNA FISH GAPDH exon/intron 2 overlap. GAPDH transcription sites were decreased in the majority of cells after 40 min of treatment, while after 2.5 and 4.5 h the signal was not detectable (Supplementary Fig. 8). Collectively, these results show that intron-containing TUG1 and TERT are stable, long-lived transcripts, stably retained in the nucleus relative to their spliced cytoplasmic counterparts. We note that nuclear RNAs with retained introns were shown to be hyperadenylated upon transcription inhibition, which may contribute to their stabilization[59].

**TERT pre-mRNA splicing is cell-cycle specific occurring at mitosis.** During our smRNA FISH analyses of intron 11 retention, we observed that TERT pre-mRNA was spliced during cell division (after late prophase). Specifically, we used DNA staining with Hoechst to distinguish cells in interphase and mitosis (either prophase, metaphase, anaphase or telophase). We quantified unspliced TERT (co-localized intron 11 and exon signal), spliced

TERT (exon signal only), and solo intron 11 in each stage after late prophase. In contrast to interphase, during mitosis, all TERT RNA molecules could be readily visualized as spliced. We observed that TERT intron 11 was in the form of a solo intron, i.e., not co-localized with exons (Fig. 6a, b). The quantity of spliced TERT was increased in mitosis compared to interphase cells (mean value 9.4 vs. 3.8 of spliced TERT molecules per mitotic or interphase cell, respectively, $P \leq 0.001$, unpaired $t$-test). Moreover, the quantity of unspliced TERT was reduced from a mean value of 10.4 molecules/cell in interphase cells to 1.0 in mitosis ($P \leq 0.001$, unpaired $t$-test). Lastly, while intron 11 was included in the vast majority of nuclear TERT mRNA in interphase cells, in mitosis the quantity of free intron 11 greatly increased (mean value 0.0 vs. 8.2, respectively, $P \leq 0.001$, unpaired $t$-test).

Notably, the quantity of free intron 11 was comparable with the number of newly spliced TERT RNA molecules (mean value 8.2 vs. 9.4, respectively). Since the mitotically spliced intron was observed by smRNA FISH, it further indicates that the intron was stable, presumably in the form of a lariat. However, given that the free intron was not observed in other stages of the cell cycle, neither in the cytoplasm nor in the nucleus, it indicates that the stability of solo intron 11 is limited to mitosis.

We performed the same analysis for TUG1. In contrast to TERT, a great portion of TUG1 transcripts remained unspliced during mitosis compared to interphase cells (Fig. 6c, d, mean value 10.6 vs. 16.4 for Δintron1, respectively; 10.9 vs. 14.9 for Δintron2, respectively), implying that TUG1 splicing is not dependent on mitosis. Together, our smRNA FISH analysis found that TERT splicing of retained intron 11 appears to be regulated in mitosis, opening an intriguing possibility of mitotic inheritance of fully spliced, cytoplasmic TERT mRNA.

**Modified ASOs block splicing and affect subcellular RNA localization.** All our observations above of nuclear TERT and TUG1 intron retention are correlative and do not show a causality of intron retention driving their subcellular localization. To test the causality, we applied chemically modified ASOs called Thiomorpholinos (TMOs). TMOs are oligonucleotides in which the bases (thymine, cytosine, adenine, and guanine) are attached to morpholine, and these nucleosides are joined through thiophosphoramidate internucleotide linkages (Fig. 7a)[60]. They show increased hybridization stability towards complementary RNA (10 °C increased melting temperature compared to an unmodified control duplex of identical sequence). TMOs are also highly stable towards exonuclease enzymes; minimal degradation is observed when treated with snake venom phosphodiesterase I for over

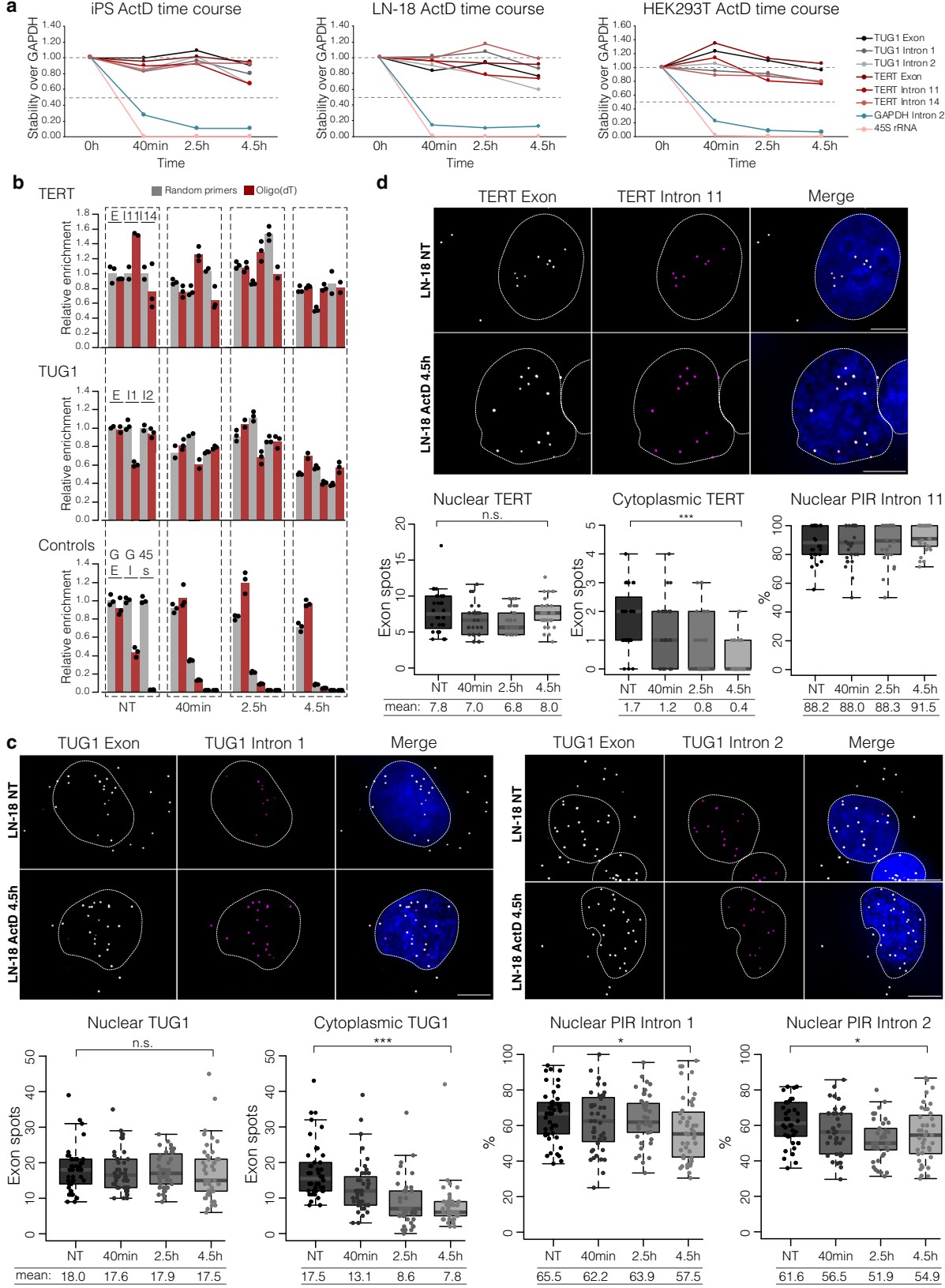

23 h. Unlike DNA:RNA duplexes, they do not elicit RNase H1 activity, making them ideal candidates for splicing studies.

We hypothesized that increasing intron retention via blocking excision of the retained introns would affect the subcellular localization of TUG1. We further wanted to test the physiological role of TUG1 subcellular localization in cancer cell lines, as it has been implicated in many cancers. Specifically, we chose U-2 OS and HeLa for TUG1 lncRNA based on the previous studies[48,49,61]. To test this hypothesis, we designed 20-mer TMOs against the two TUG1 donor splice sites, each hybridizing to 2 nt

**Fig. 5 Intron-retained nuclear TUG1 and TERT are long-lived transcripts, stably retained in the nucleus. a** Relative stability of TUG1 and TERT exons and introns compared to GAPDH mRNA measured by RT-qPCR of random-primed cDNA from iPS, HEK293T and LN-18 cells during a 4.5 h ActD time course. GAPDH intron 2, a control for an efficiently spliced intron; 45S rRNA, a control for a precursor RNA. Each dot represents mean value from two or three replicates. **b** Relative abundance and stability of spliced and intron-retained transcripts in cDNA synthesized with random primers or oligo(dT) during the 4.5 h ActD treatment of iPS cells; bars, means across replicates; dots, individual replicates, n = two or three measurements. **c** Maximum intensity projection of LN-18 smRNA FISH targeting TUG1 exon (gray) and intron 1 (magenta) or intron 2 (magenta) at time point 0 (NT) and 4.5 h after ActD treatment. Scale bar, 5 µm. Below, smRNA FISH quantification at each time point of spliced and unspliced TUG1 transcripts in the nucleus and cytoplasm; PIR of intron 1 and intron 2 at each time point. n.s. = not significant, *P ≤ 0.05, ***P ≤ 0.001, evaluated by unpaired two-tailed *t*-test (equal variances) versus NT; n (nuclear TUG1, cytoplasmic TUG1, nuclear PIR intron 1) = 44 cells (NT, 40 min, 4.5 h), 43 cells (2.5 h); n (nuclear PIR intron 2) = 38 cells (NT), 39 cells (40 min, 4.5 h), 40 cells (2.5 h), two independent RNA FISH stainings. **d** Maximum intensity projection of LN-18 smRNA FISH targeting TERT exon (gray) and intron 11 (magenta) at time point 0 (NT) and 4.5 h after ActD treatment. Scale bar, 5 µm. Below, smRNA FISH quantification at each time point of spliced and unspliced TERT transcripts in the nucleus and cytoplasm; nuclear PIR of intron 11 at each time point. n.s. = not significant, ***P ≤ 0.001, evaluated by unpaired two-tailed *t*-test (equal variances) versus NT, n (nuclear, cytoplasmic TERT, nuclear PIR intron 11) = 32 cells (NT), 30 cells (40 min, 2.5 h, 4.5 h), two independent RNA FISH stainings. In **c**, **d** midline line, median; lower and upper box limits, 25th and 75th percentiles; whiskers, 1.5 times interquartile range from the 25th and 75th percentiles.

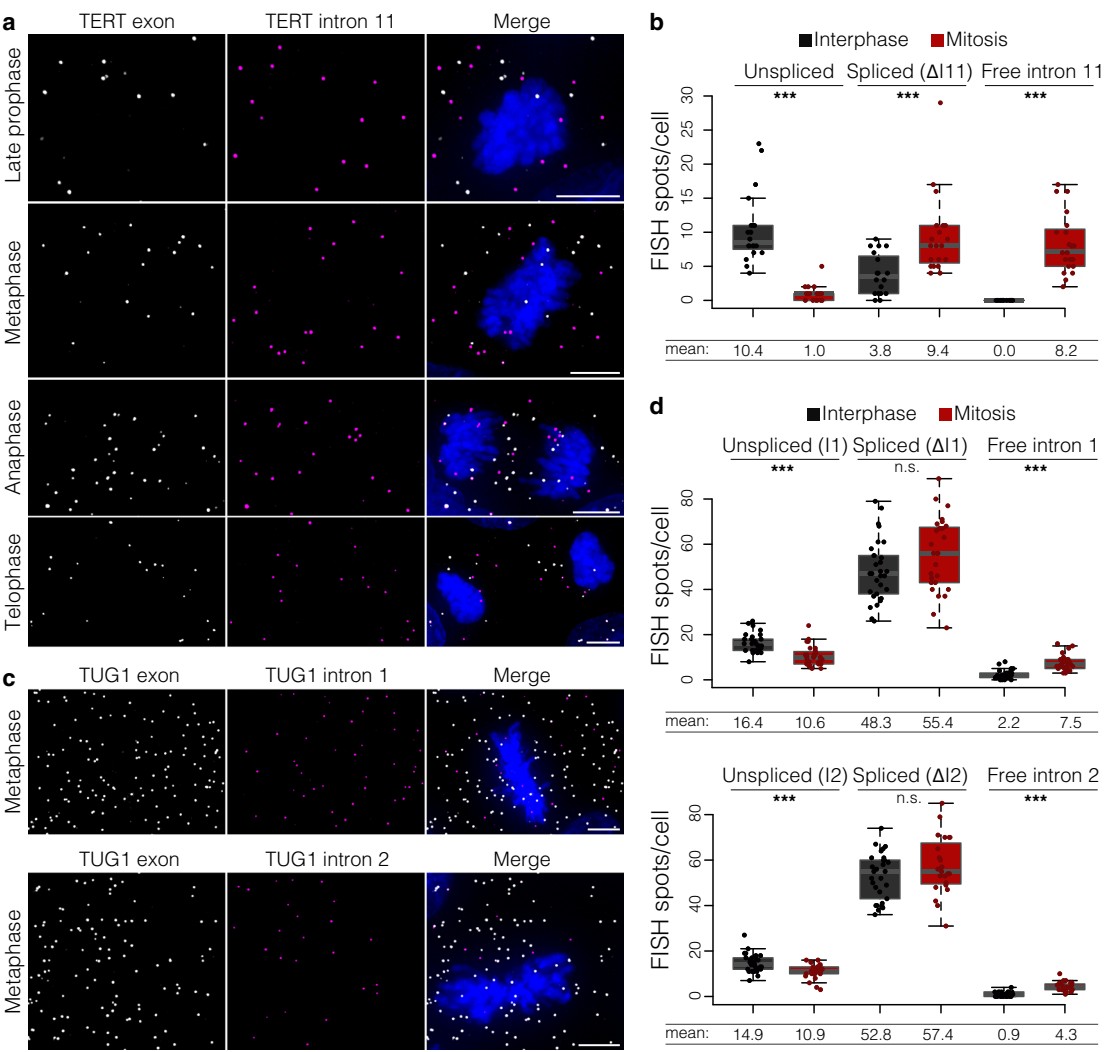

**Fig. 6 Splicing of TERT intron 11 occurs upon mitosis. a** Maximum intensity projections of TERT exon (gray) and intron 11 (magenta) smRNA FISH. Representative images of late prophase, metaphase, anaphase, and telophase are shown. DAPI shown in blue. Scale bar, 5 µm. Three independent experiments. **b** Quantification of unspliced TERT, spliced (ΔI11) TERT, and free intron 11 in interphase cells and during mitosis. ***P ≤ 0.001, evaluated by unpaired two-tailed *t*-test (equal variances) versus interphase; n = 20 cells from three independent RNA FISH stainings. **c** Maximum intensity projections of TUG1 exon (gray) and intron 1 (magenta) or intron 2 (magenta) smRNA FISH. Representative images of metaphases are shown. DAPI shown in blue. Scale bar, 5 µm. Two independent experiments. **d**, Quantification of unspliced TUG1, spliced (ΔI1 or ΔI2) TUG1, and free intron 1 or 2 in interphase cells and during mitosis. n.s. = not significant, ***P ≤ 0.001, as evaluated by unpaired two-tailed *t*-test (equal variances) versus interphase; for intron 1 n (interphase) = 30 cells, n (mitosis) = 27 cells; for intron 2 n (interphase) = 29 cells, n (mitosis) = 23 cells; two independent RNA FISH stainings. In **b**, **d** midline line, median; lower and upper box limits, 25th and 75th percentiles; whiskers, 1.5 times interquartile range from the 25th and 75th percentiles.

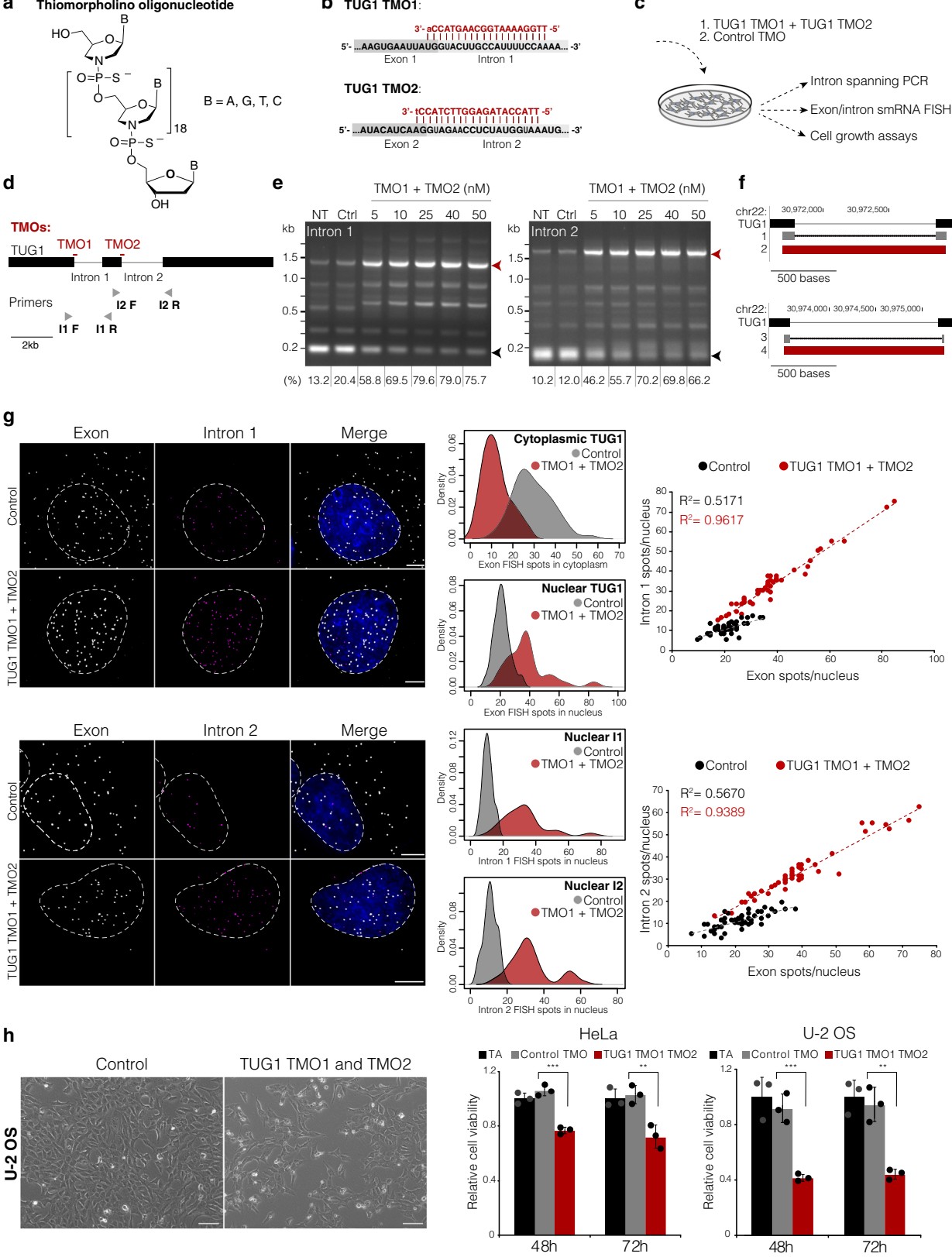

of the exon and 18 nt of the intron sequence (designated TUG1 TMO1 and TMO2) (Fig. 7b). To control for cell effects that could be caused by TMO intake, we designed a control TMO (randomized sequence of TMO1).

TMOs were transfected at an increasing concentration to U-2 OS and HeLa cells, after which cells were harvested for monitoring intron retention via intron-spanning RT-qPCR and smRNA FISH (Fig. 7c, d). The mixture of TUG1 TMO1 and

**Fig. 7 Intron retention drives nuclear compartmentalization of TUG1. a** The chemical structure of thiomorpholino oligonucleotide (TMO). **b** The design of TUG1 TMO1 and TMO2 (in red) against the donor splice sites. For TMOs, upper-case red letters refer to thiomorpholino nucleotides and lower-case letters to 2′-deoxynucleosides at the 3′ end of each TMO. **c** Experimental setup to assess the efficiency of TMO-based intron inclusion and its effect of subcellular localization of TUG1 and cell viability. **d** TMO location scheme in respect to TUG1 transcript and the location on intron-spanning primers (not to scale). **e** PCR product of the intron-spanning RT-PCR of untreated (NT), control TMO (Ctrl), and increasing doses of a mixture of TUG1 TMO1 and TMO2. Black arrow, spliced product; red arrow, unspliced product. Below, the percentage of the unspliced products. Kb, kilobases. PCR products after transfecting TUG1 TMOs were examined on agarose gel at least three independent times. Uncropped blot is provided in Source data. **f** UCSC browser displaying Sanger sequencing results of spliced (band 1) and unspliced (band 2) products for intron 1 RT-PCR (on top). Below, the sequences for spliced (band 3) and unspliced (band 4) products for intron 2 RT-PCR. **g** Maximum intensity projections of TUG1 exon and intron 1 or intron 2 smRNA FISH in U-2 OS cells transfected with control TMO and with TUG1 TMO1 and TMO2. Exon, gray; intron, magenta; nucleus, blue; scale bar, 5 μm. Towards the right, quantification of nuclear TUG1, cytoplasmic TUG1, intron 1 or 2 retentions in TUG1 TMO1 and TMO2 (red) versus control TMO (gray) samples, $n = 50$ cells (control), 49 cells (intron 1 TMO1 and 2), 44 cells (intron 2 TMO1 and 2). **h**, Relative cell viability of HeLa and U-2 OS cells transfected with TUG1 TMO1 and TMO2, control TMO or transfection agent only (TA). Representative images of U-2 OS transfected with control TMO or TUG1 TMO1 and TMO2 shown on the left. Scale bar, 100 μm. $**P \leq 0.01$, $***P \leq 0.001$, as evaluated by unpaired two-tailed $t$-test (equal variances) versus control TMO. Bars, means across replicates; dots, individual replicates, error bars, the standard deviation of the mean of three independent measurements.

TMO2 inhibited splicing and achieved retention of both introns in a dose-dependent manner already 24 h after treatment (Fig. 7e). Sanger sequencing of the spliced and unspliced RT-PCR products confirmed that the complete introns were retained (Fig. 7f). Interestingly, alongside the expected spliced and intron-retained amplicons, the intron-spanning PCR revealed additional isoforms amplified by the specified primer sets. Some of those less abundant isoforms displayed a shift in size upon the TMO treatment (Supplementary Fig. 9a, labeled with a star).

We next used smRNA FISH to determine whether forced intron inclusion would affect the subcellular localization and availability of spliced TUG1 in the cytoplasm. Specifically, we performed dual-color smRNA FISH in U-2 OS and HeLa cell lines treated with TUG1-targeting TMOs (TUG1 TMO1 + TMO2) and a control TMO. The TUG1-targeting TMOs gave a drastic shift in the subcellular localization and splicing of TUG1 (Fig. 7g and Supplementary Fig. 9b). On average, in U-2 OS TUG1 decreased ~2.4-fold in the cytoplasm (mean 29 molecules in control vs. 12 in TUG1 TMO1 + 2), while TUG1 increased ~1.8-fold in the nucleus (mean 21 molecules in control vs. 38 in TUG1 TMO1 + 2). Similarly, in HeLa cells, TUG1 decreased ~2.7-fold in the cytoplasm (mean 22 molecules in control vs. 8 in TUG1 TMO1 + 2), and it increased ~1.7-fold in the nucleus (mean count 29 in control vs. 48 in TUG1 TMO1 + 2) (Fig. 7g and Supplementary Fig. 9b). After TUG1 TMO1 + TMO2 application, intron retention in nuclear TUG1 was significantly increased in U-2 OS and HeLa cells (PIR intron 1 increased from 51 to 85%, PIR intron 2 increased from 52 to 84% in U2-OS; PIR intron 1 increased from 67 to 92%, PIR intron 2 increased from 57 to 92% in HeLa).

In parallel, we used TUG1 TMO1 labeled with FITC (TUG1 TMO1-FITC) to determine the subcellular localization of the TMO after transfection. TUG1 TMO1-FITC showed predominantly nuclear localization, and it was stably localized in the nucleus 96 h after transfection (later time points were not assessed) (Supplementary Fig. 9c), consistent with these oligos being able to alter nuclear splicing processes. Together, our results demonstrate that TMOs can be used to achieve increased intron retention and in turn increased nuclear localization of transcripts.

**Functional consequences of enforced intron retention for TUG1 and TERT.** Having observed that increasing intron retention increases nuclear localization of TUG1, we wanted to determine if this redistribution of transcripts has a functional cellular consequence. Specifically, U-2 OS and HeLa cell lines were transfected with 25 nM of TUG1 TMO1 and TMO2 and cell viability was assessed 48 h and 72 h post-transfection relative to transfection agent only and a control TMO (Fig. 7h). Both cell

lines showed a reduction in cell viability after 48 h of TUG1 TMO treatment compared to controls (mean 24% and 59% reduced viability for HeLa and U-2 OS, respectively, $P \leq 0.01$, unpaired $t$-test), and after 72 h (mean 29% and 57% reduced viability for HeLa and U-2 OS, respectively, $P \leq 0.01$, unpaired $t$-test). Thus, in both cases altering the subcellular distribution of TUG1 impaired cell viability.

To determine if our TMO strategy is also applicable to pre-mRNAs we focused on TERT. Briefly, we designed a TMO to retain specified intron 11 of TERT targeting TERT exon 11/ intron 11 junctions and determined the cellular consequences thereof (Fig. 8a). Cell lines with uniform reactivation of TERT expression, LN-18, and HEK293T, were transfected with TERT TMO and control TMO. Because TERT intron 11 is quite long (3.8 kb), intron-spanning PCR was not feasible to assess intron retention. Thus, we applied exon/intron junction RT-qPCR to assess the efficiency of intron retention (Fig. 8b, c). We found that TMOs enforcing intron 11 retention decreased splicing of intron 11 by ~60% compared to the control TMO. In contrast, intron 11-containing TERT (assessed by monitoring intron 11 and exon 11 to intron 11 junction) was increased ~32% compared to control TMO. As additional controls, we applied primers at the upstream exon 10 to exon 11 junction, which was not affected with TERT TMO treatment, and exon 10 to exon 12 junction, which was decreased ~50%, in accordance with the decrease in exon 11 to exon 12 junction (Fig. 8c).

We further leveraged the specific retention of intron 11 and the restriction of splicing to mitosis. We observed that the total number of TERT RNA molecules (assessed by overall exon signal) was not altered during mitosis between control and TERT TMO-treated samples (mean values 6.3 and 6.7, respectively). Consistent with the above results, we observed a significant effect of TERT TMO on splicing of intron 11 only during mitosis (Fig. 8d). More specifically, the majority of intron 11 was spliced out and observed in the form of a solo intron with control TMO, and these solo introns were significantly decreased in cells treated with TERT TMO (mean value 4.0 solo intron 11 per mitosis in control TMO and 1.3 solo intron 11 per mitosis in TERT TMO, $P \leq 0.001$, unpaired $t$-test).

The observation of solo intron 11 during mitosis is in accordance with the observation made in iPS cells (Fig. 6a, b). While during mitosis most of TERT RNA was in the form of spliced RNA in the control TMO samples, the quantity of spliced TERT significantly decreased in TERT TMO samples (mean values 5.4 and 2.2 molecules per mitosis, respectively, $P \leq 0.001$, unpaired $t$-test). Consequently, the quantity of unspliced TERT increased in TERT TMO compared to control TMO samples (mean values 4.5 and 0.8 molecules per mitosis, respectively, $P \leq$

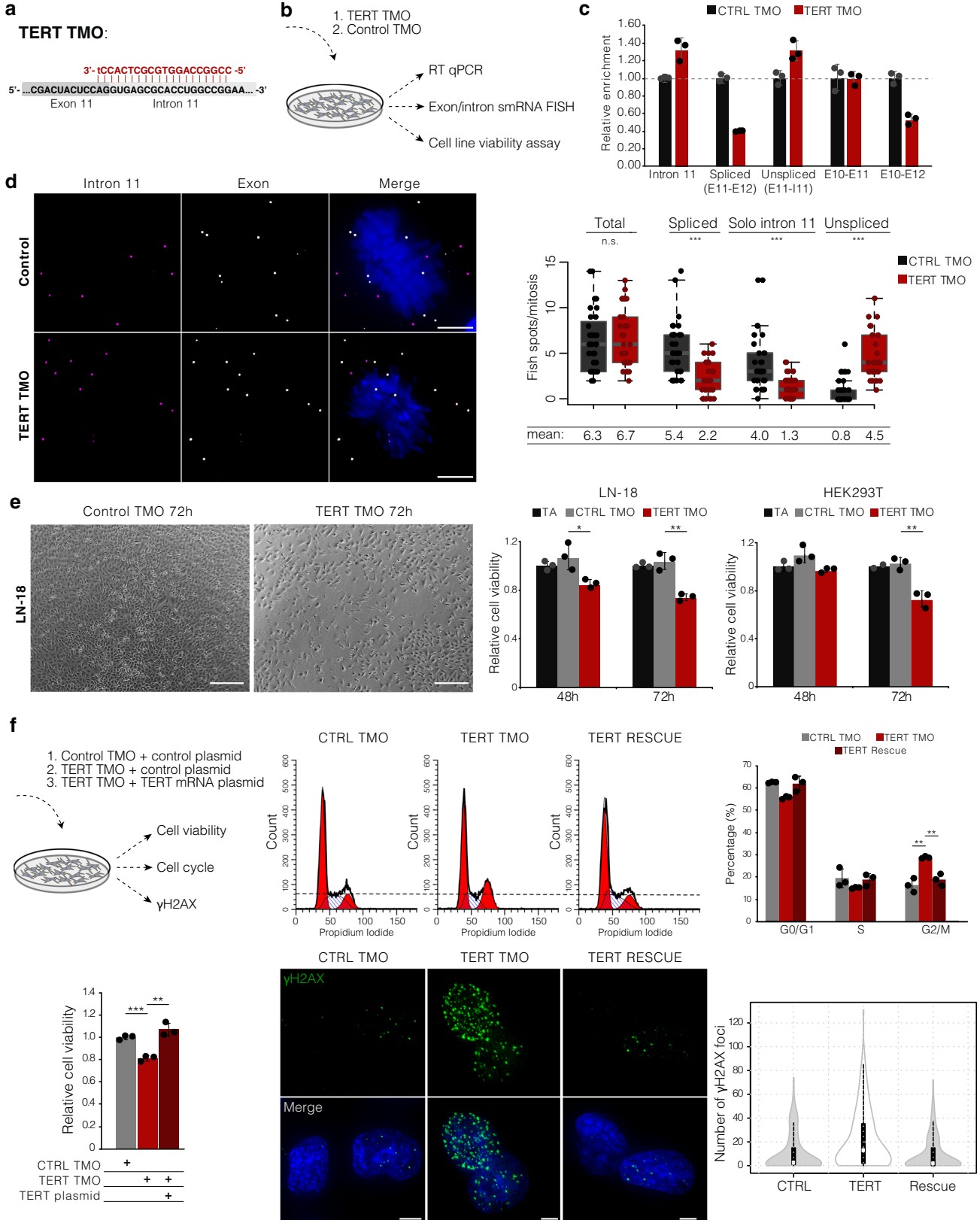

0.001, unpaired *t*-test). Overall, the RT-qPCR and smRNA FISH confirmed that TMOs can specifically inhibit splicing of intron 11 from TERT pre-mRNA.

We next sought to determine whether inhibiting the availability of spliced TERT by TERT TMO would affect cell growth of

LN-18 and HEK293T cell lines. Both cell lines were transfected with 25 nM of TERT TMO, and cell proliferation was assessed (Fig. 8e). LN-18 cell line showed a reduction in cell viability after 48 h of treatment compared to transfection agent only and control TMO (mean reduction of 18%, $P \leq 0.05$), which was

**Fig. 8 TMO-based prevention of TERT splicing reduces cell viability in vitro. a** Scheme showing the design of TERT TMO (in red) against the exon11/intron11 donor splice site. The upper-case red letters refer to thiomorpholino nucleotides and the lower-case letter to a 2′-deoxynucleoside at the 3′ end. **b** Experimental setup to assess the efficiency of TMO-based TERT intron 11 inclusion (RT-qPCR and smRNA FISH) and its effect on cell viability. **c** Relative expression of TERT intron 11, spliced TERT (Exon10-Exon11, Exon10-Exon12, Exon11-Exon12), and unspliced TERT (Exon11-Intron11) over GAPDH assessed by RT-qPCR. Error bars represent the standard deviation of the mean of three replicates. **d** Maximum intensity projections of TERT exon (gray) and intron 11 (magenta) smRNA FISH in LN-18 cells transfected with control TMO and TERT TMO. DAPI, blue. Scale bar, 5 μm. On the right, quantification of total TERT (exon signal), unspliced TERT, spliced TERT (ΔI11), and free intron 11 during mitosis of LN-18 cells transfected with control TMO (CTRL) or TERT TMO. N (control TMO) = 32 cells, n (TERT TMO) = 30 cells, two independent RNA FISH stainings. Midline line, median; lower and upper box limits, 25th and 75th percentiles; whiskers, 1.5 times interquartile range from the 25th and 75th percentiles. **e** Cell viability of LN-18 and HEK293T cells transfected with TERT TMO, control TMO or transfection agent only (TA). Representative images of LN-18 transfected with control TMO or TERT TMO shown on the left. Scale bar, 250 μm. Bars, means across replicates; dots, individual replicates, error bars, standard deviation of the mean of three independent measurements. **f** Cell viability, cell cycle, and γH2A.X foci analysis of LN-18 cell line 72 h after co-transfection of TERT TMO with spliced TERT expression plasmid, control TMO, and TERT TMO with a control plasmid. Cell viability and cell cycle; bars, means across replicates; dots, individual replicates; error bars, standard deviation of the mean of three independent measurements. Representative images of γH2A.X immunofluorescence are shown, DAPI, blue; γH2A.X, green; scale bar 5 μm. Violin plot, n (control TMO) = 128 cells, n (TERT TMO) = 168 cells, n (rescue) = 153 cells, two independent stainings. White circles, median; box limits indicate the 25th and 75th percentiles; whiskers, 1.5 times the interquartile range from the 25th and 75th percentiles; polygons represent density estimates of data and extend to extreme values. For **d**, **e**, and **f** p values were obtained by unpaired two-tailed t-test (equal variances), n.s. = not significant, *$P \leq 0.05$, **$P \leq 0.01$, ***$P \leq 0.001$.

further enhanced after 72 h (mean reduction of 28% in cell viability, $P \leq 0.01$). HEK293T showed a delayed response and cell viability was reduced after 72 h treatment with TERT TMO (mean reduction of 29% in cell viability, $P \leq 0.01$) compared to control TMO and transfection agent only.

It is intriguing that cell viability is compromised so quickly after reduction of translatable TERT mRNA, given that telomere shrinkage due to inhibition of telomerase typically takes many population doublings before it gives a growth defect[62]. Indeed, we detected only mild telomere shortening after 5 days of treatment with TERT TMO by Southern blot (Supplementary Fig. 10a). This result coincides with previous cases where telomerase inhibition was found to decrease cell viability through telomere length-independent processes[63–66]. More specifically, we observed an induction of γH2A.X foci and G2/M arrest as has been shown to occur upon short-term telomerase inhibition (Supplementary Fig. 10b, c). To test the specificity of the TERT TMO effect on cell viability, we co-transfected spliced TERT (therefore, not a target of the TERT TMO) in combination with TERT TMO. Effects specific to inhibiting excision of intron 11 with TERT TMO, reduction of cell viability, G2/M arrest, and γH2A.X foci, were absent when co-transfecting spliced TERT, supporting the specificity of these effects (Fig. 8f).

Collectively, these findings demonstrate that TMOs effectively block splicing and change cellular localization and availability of the RNA. Moreover, we find that this perturbed subcellular transcript distribution has a functional consequence on cell viability.

## Discussion

It has long been known that the spatio-temporal distribution and compartmentalization of RNA in the cell is tightly coupled with its subcellular function[1–6]. Studies of underlying mechanisms have pinpointed RNA motifs and structural features that target RNA subcellular localization[67,68]. Splicing has been shown to strongly influence RNA localization[9,10]. For example, lncRNAs are inefficiently spliced, display increased intron retention relative to coding mRNAs, and are more nuclear than mRNAs[11,55,56]. Despite these intriguing findings, the causality of splicing events, such as intron retention, in the molecular events driving nuclear retention of RNAs has remained unclear. A surprising finding is that the majority of TERT transcripts are nuclear, and therefore translationally inert, yet the underlying mechanism remained unknown[39,40].

Here we addressed this question using multiple approaches including single-molecule RNA FISH to spatio-temporally measure specific splicing events that may alter subcellular RNA localization. We focused on two cancer-related transcripts, TERT mRNA and TUG1 lncRNA. We find that both mRNA and lncRNA localization patterns are driven by consistent retention of specific introns. While such RNAs could be nonfunctional, partially spliced byproducts, or serve as nuclear non-coding RNAs, it has been shown that some of the nuclear, intron-retained RNAs are poised or 'detained' for a signal for post-transcriptional splicing, hence serving as a reservoir of RNAs readily available depending on cellular activity[12,13]. In this regard, the striking splicing of retained TERT intron 11 after cells' entry to mitosis was an intriguing indication that fully spliced TERT might be generated mitotically. Retention of specific introns would compartmentalize TERT RNA in the nucleus of interphase cells, while upon cells' entry to mitosis, retained introns would be spliced out and daughter cells would inherit fully spliced TERT. Mitotic inheritance of spliced TERT would ensure that telomere elongation occurs only in mitotically active cells, still allowing telomerase assembly during the later stages of the cell cycle when DNA is replicated and telomeres elongated[69–71].

Together, these findings raise the question of how intron 11 is retained in order to specifically be spliced out during mitosis and produce a cytoplasmic transcript for translation. In this regard, it is interesting to consider that TERT intron retention may be regulated as part of a broader program of differential intron retention (and other forms of alternative splicing) that is controlled by the SR protein splicing factor kinase CLK1 during the cell cycle[72]. Alternatively, possibly some other signaling pathway could regulate the splicing of the retained introns and nuclear export of TERT for translation during interphase.

We found that the lncRNA TUG1 is equally distributed between the nucleus and cytoplasm across multiple cell lines. Hence the same locus gives rise to equal amounts of either efficiently spliced cytoplasmic TUG1 or intron-retained nuclear TUG1, where intron retention dictates nuclear/cytoplasmic transcript distribution. This interesting splicing balance could have important implications: (i) the longer TUG1 lncRNA with retained introns could exert a specific nuclear RNA function in this longer form, which is consistent with the strong conservation of TUG1 intronic sequences—an infrequent property of lncRNA or mRNA introns; (ii) intron sequences could give rise to distinct functions; (iii) the efficiently spliced cytoplasmic TUG1 could be destined to encode a protein, as has been proposed by recent

studies[51]; (iv) the conserved distribution of TUG1 in the nucleus and cytoplasm could represent a translational buffering or two distinct functionalities. One of these mechanisms, or their combination, potentially underlies a 100% penetrant male infertility phenotype in TUG1 knock-out mouse models[51].

Both TERT and TUG1 are upregulated in many cancers and thus represent important therapeutic targets. To this end, we tested a RNA-based strategy to alter TERT and TUG1 splicing and subcellular distribution. We found that this TMO antisense approach was highly effective and specific at blocking TERT and TUG1 splicing events. Importantly, altering these specific splicing patterns using our TMO approach not only affected subcellular distribution but, in both cases, decreased cell viability. Thus, TMO-based strategies could be universally applicable not only to other transcripts that retain specific introns, but to a variety of oncogene transcripts that could be rendered inert in the nucleus.

## Methods

**Cell lines and cell culture**. HCT116 (CCL-247), HeLa (CCL-2), HEK293T (CRL-3216), LN-18 (CRL-2610), U-2 OS (HTB-96) cell lines were obtained from ATCC and cultured according to recommended protocols. Human iPSC WTC-11 (Coriell Institute) cells were cultured on Vitronectin (Thermo Fisher Scientific) coated 6-well plates or glass coverslips (for smRNA FISH purposes) in Essential 8 Flex medium (Thermo Fisher Scientific) with E8 supplement (Thermo Fisher Scientific), Rock inhibitor and 2.5% penicillin-streptomycin. iPS cells we passaged with EDTA in dPBS. Mouse embryonic stem cells (Harvard Stem cell institute) were cultured on top of gelatin (0.1%, EMD Millipore) coated plates or glass coverslips (for smRNA FISH purposes). Embryonic stem cell media was prepared as follows: KnockOut DMEM medium (Thermo Fisher) supplemented with ESC FCS (Millipore Sigma), non-essential amino acids (Thermo Fisher), GlutaMAX supplement (Thermo Fisher), penicillin-streptomycin (Thermo Fisher), 50 mM 2-mercaptoethanol, LIF, CHIR99021 and PD0325901 (Sigma-Aldrich).

Actinomycin D (Sigma-Aldrich) was used at final concentration of 5 μg/mL in full growth media. Cell pellets and coverslips were harvested at 0, 40 min, 2.5 h and 4.5 h after adding Actinomycin D, and processed for RNA extraction and smRNA FISH as described below.

**RNA extraction**. After the corresponding treatments, cell pellets were harvested and RNA extraction was performed with Maxwell LEV Simply RNA tissue kit (Promega) following manufacturer's instructions with DNase I treatment. Each sample was tested for DNA contamination by qPCR after each extraction. RNA quality was assessed on 2% agarose gel and Bioanalyzer (RNA Nano Assay: 25–500 ng/μL).

**Analysis of intron retention from RNA-seq**. Vast-tools v2.2.2[54] (https://github.com/vastgroup/vast-tools) was used to calculate PIR values. We used two previously published poly(A)+ RNA-seq data from human iPS cells[73] (GSM1023087 and GSM1023070), and one ribosomal RNA depleted RNA-seq from human iPS cells (GSM808734). For mouse, we used two previously published poly(A)+ RNA-seq datasets from mouse iPS cells (GSM1032506, GSM1032518) and three poly(A)+ RNA-seq datasets from mES cells generated in this study and available from GSE169743.

Reads mapping to mid-intron sequences and balanced counts of reads aligning to upstream and downstream exon–intron sequences were used to evaluate intron retention levels. PIR was measured as a percentage of mean retention reads over the sum of retained and spliced intron reads. Raw values were filtered based on reported quality scores, requiring at least 15 total reads per event and absence of a positive result ($P < 0.05$) for the binomial test for upstream/downstream junction read balance. PIR values for human TERT intron 11 were reported by vast-tools as imbalanced due to an alternative exon within the intron and were therefore re-calculated based solely on the downstream intron-exon junction reads. Similarly, TUG1 intron 1 was re-calculated based on upstream exon–intron junction reads due to an alternative acceptor site, and TUG1 intron 2 was absent from the VastDB database. For the analysis of global levels of maximum and minimum PIR in coding and non-coding genes, gene biotype annotations were taken from GENCODE v29 (human) and vM23 (mouse) and simplified to 'coding', 'lncRNA', and 'other' (not shown).

**Methods for analysis of splicing conservation in TERT**. Vast-tools results from poly(A)+ RNA-seq datasets for introns in TERT in seven mammalian species across multiple tissue and cell line samples (*H. sapiens*, 128; *P. troglodytes*, 38; *M. mulatta*, 66; *R. norvegicus*, 127; *M. musculus*, 151; *B. taurus*, 58; *M. domestica*, 44) were kindly shared by M. Irimia (CRG, Barcelona). Vast-tools results were filtered as described for RNA-seq analysis in iPS cells.

**Analysis of RBPs interactions in introns**. eCLIP peak bed files for 223 experiments in HEPG2 and K562 cell lines were retrieved from the ENCODE Data Portal and merged for each RBP using GenomicRanges (R, version 3.6.0; Bioconductor, version 3.10). Intron windows for the primary transcript of TERT (ENST00000310581.9) were generated by taking 40 bp upstream and downstream of the 3′ and 5′ splice sites and the remaining interior of the intron was partitioned into five equal tiles. The fraction of base pairs in each window was visualized using ComplexHeatmap[74]. For the RBP motif analysis, motifs were retrieved from the ATtRACT motif database and converted to MEME format using chen2meme from the MEME suite (version 5.1.1)[75,76]. FIMO from the MEME suite was used to search for motif matches in the intronic sequence and a *p*-value cut-off of $P < 1e-4$ was used to filter low-quality motif instances. Since multiple motifs can refer to the same RBP, for ease of visualization motif matches were collated by RBP and visualized in a heatmap using the same windows as for eCLIP. The color in the motif heatmap corresponds to the maximum FIMO score within that window for a given RBP.

**cDNA synthesis and qPCR analysis**. Reverse transcription was performed with SuperScript® IV First-Strand Synthesis System (Thermo Fisher Scientific) with Superase RNase inhibitor (Ambion) and random hexamers hexamers or oligo dT (indicated in the figure legend) on 0.2–1 μg of RNA. Relative expression was determined by qPCR using SYBR Green I master mix (Thermo Fisher Scientific) according to manufacturer's instructions using the following amplification conditions: 95 °C 10′; 45 cycles of 95 °C 15″, 57.5 °C 20″ and 72 °C 25″. Expression levels were normalized using GAPDH. A list of primers used in qPCR analyses are summarized in Supplementary Table 1. Their efficiencies were compared to ensure analysis by the comparative Ct method. Relative expression data were analyzed comparing the Ct values of the gene of interest with Ct values of the reference gene for every sample. We used the formula $2\Delta\Delta Ct$, $\Delta\Delta Ct$ being the difference between the Ct of the RNA of interest and the Ct of the housekeeping gene. Duplicates or triplicates were made for each sample and primer set.

**RT-PCR and Sanger sequencing**. cDNA was amplified with Q5® High-Fidelity DNA Polymerase PCR System (NEB). PCR conditions: initial denaturing at 95 °C 2′; 40 cycles of denaturing at 95 °C 30″, annealing at 58 °C 30″ and extension at 72 °C 2′ 30″; followed by final extension at 72 °C 7′. PCR product was examined on a 1% agarose gel for correct size and specificity. Bands corresponding to spliced or unspliced TUG1 were cut from the gel and DNA was extracted with Gel extraction kit (Qiagen) according to instructions. Extracted DNA was cloned with TOPO PCR Cloning Kit (Thermo Fisher Scientific), and positive colonies selected on ampicillin agar plates. Minipreps from ∼5 colonies for each amplicon were sent for Sanger sequencing to Genewiz using T3 or T7 primers.

**Single-molecule RNA FISH**. For in situ RNA detection, single-molecule RNA FISH was employed[30]. Tiled oligonucleotides targeting human and mouse TUG1 exons, TERT intron 2, TERT exons, GAPDH intron 2, and GAPDH exons labeled with either Quasar 570 or Quasar 670 were used in our previous studies[40,51]. For this study, we custom designed tiled oligonucleotides targeting human and mouse TUG1 intron 1 (Quasar 570) and intron 2 (Quasar 570), TERT intron 11 (Quasar 670), and TERT intron 14 (Quasar 670) using LGC Biosearch Technologies' Stellaris online RNA FISH probe designer (Stellaris Probe Designer, version 4.2) which were produced by LGC Biosearch Technologies.

Cells were seeded on glass coverslips coated with poly-L-lysine (10 μg/mL in PBS), vitronectin (human iPS cells) or gelatine (mouse ES cells). The iPS cells were seeded at lower density the day before harvesting the coverslips to facilitate the quantification process. Coverslips were washed 2 times with PBS, fixed in 3.7% formaldehyde in PBS for 10 min at room temperature (RT), followed by washing 2 times with PBS and immersed in 70% EtOH at 4 °C for a minimum of 1 h. Prior hybridization, coverslips were washed with 2 mL of wash buffer A (LGC Biosearch Technologies) supplemented with 10% deionized formamide (Agilent) at RT for 5 min. Cells were hybridized with 80 μL of hybridization buffer (LGC Biosearch Technologies) supplemented with 10% deionized formamide (Agilent) containing 1:100 dilution of smRNA FISH probes overnight at 37 °C in a humid chamber. The next day, cells were washed with 1 mL of wash buffer A with 10% formamide for 30 min at 37 °C, followed by a wash with wash buffer A with 10% formamide containing Hoechst DNA stain (1:1000; Thermo Fisher Scientific) for 30 min at 37 °C. Coverslips were washed with 1 mL of wash buffer B (LGC Biosearch Technologies) for 5 min at RT, equilibrated 5 min in base glucose buffer (2× SSC, 0.4% glucose solution, 20 mM Tris pH 8.0 in RNase-free H₂O), and then incubated 5 min in Base Glucose buffer supplemented with 1:100 dilution of glucose oxidase (stock 3.7 mg/mL) and catalase (stock 4 mg/mL). Afterwards, the coverslips were mounted with ProlongGold or ProlongGlass (Life Technologies) on a glass slide and left to curate overnight before proceeding to image acquisition (see below).

**Immunofluorescence**. Cells were seeded on top of poly-L-lysine coated glass coverslips. After treatment, coverslips were rinsed 2× with PBS, crosslinked in 3.7% formaldehyde in PBS, washed 3× for 5 min in PBS/0.1% Triton-X. Cells were blocked with PBS/0.1% Triton-X/3% BSA for 30 min at RT. Cells were incubated with primary antibody against phospho-Histone H2A.X (1:100, Millipore Sigma) in

blocking solution and washed 3× for 5 min in PBS. Secondary antibody (anti-mouse labeled with Alexa Fluor 488, 1:800 in blocking solution, ab150113, Abcam) was incubated for 1 h at RT. Cells were washed 3× for 5 min in PBS, 15 min in PBS with Hoechst DNA stain (1:1000; Thermo Fisher Scientific), rinsed with PBS and mounted with ProlongGlass (Life Technologies) on top of a glass slide.

**Microscopy and image analysis**. Z stacks with 200–250 nm z-step capturing the entire cell volume were acquired with a GE wide-field DeltaVision Elite microscope with an Olympus UPlanSApo 100×/1.40-numerical aperture oil objective lens and a PCO Edge sCMOS camera using appropriate filters. The built-in DeltaVision SoftWoRx Imaging software was used to deconvolve the three-dimensional stacks. Maximum intensity projections were generated in Fiji and subjected for quantification using Fiji. To outline cell borders, we used manual segmentation. For an accurate manual cell segmentation, the saturation of the FISH channel was increased to visualize cytoplasm contours. Overlapping cells were not quantified. The brightness and contrast of each channel was adjusted. Overlapping exon/intron spots were considered as intron-retained transcripts, while exon only transcripts as spliced transcripts. Each imaging experiment was performed at least two times quantifying at least 50 cells across two independently acquired datasets. For ActD treatment and mitosis, less cells/mitosis were quantified per treatment, as indicated in the figure legend. Analysis of z-stacked was additionally performed in 3D in Imaris to confirm that nuclear intron-retained transcripts were within the nucleus. γH2A.X foci were quantified with ImageJ plugin Spot Counter v.0.14 maintaining the same spot size between conditions.

**TMO synthesis**. Prior to thiomorpholino oligonucleotide (TMO) synthesis, appropriately protected morpholino nucleosides of adenine, guanine, thymine and cytosine and their corresponding phosphorodiamidites were synthesized as reported elsewhere[60]. All TMOs were synthesized using an Applied Biosystems Model 394 Automated DNA Synthesizer using conventional DNA synthesis reagents that were purchased from Glen Research, VA. Briefly, 1.0 μM succinyl CPG support was detritylated using 3% trichloroacetic acid in dichloromethane. The 5′-unprotected nucleoside was allowed to react with a 0.1 M solution of the appropriate morpholinonucleoside phosphorodiamidite in acetonitrile in the presence of 0.12 M 5-ethylthio-1H-tetrazole (600 s coupling time). After sulfurization using 0.05 M sulfurizing reagent II in pyridine/acetonitrile, the capping step was carried using conventional Cap Mix A (acetic anhydride/tetrahydrofuran) and Cap Mix B (1-methylimidazole in acetonitrile), completing one synthesis cycle. Multiple synthesis cycles were repeated until a TMO oligonucleotide of the desired sequence was obtained. The 5′-DMT group on the solid-support bound final oligonucleotide was not detritylated so that purification could be carried out using the DMT-On/Off procedure[77]. Cleavage and deprotection was carried out using 28% aqueous ammonia at 55 °C for 16 h. After cooling to 25 °C followed by evaporation of the ammonia mixture, the oligonucleotides were purified by ion-pair reversed phase HPLC. During this process, the total reaction mixture (after evaporation to dryness) was dissolved in 3% aqueous acetonitrile and injected into an Agilent 1100 HPLC equipped with a manual injector. Due to the lipophilicity of the DMT handle, the DMT-On TMO oligonucleotide could be easily separated from failure products using a gradient of 50 mM Triethylammonium bicarbonate in acetonitrile (Agilent Zorbax C18 column, 2.0 mL flow rate). The DMT-On fractions were pooled, evaporated to dryness and treated with 50% aqueous acetic acid for 5 min. After quenching with triethylamine, the mixture was evaporated to dryness. The resulting solids were dissolved in 3% aqueous acetonitrile and the deprotected TMO oligonucleotides were re-purified by ion-pair RP-HPLC. All oligonucleotides were desalted prior to use. Graphical illustration of thiomorpholino oligonucleotide synthesis shown in Supplementary Fig. 11.

**TMO and plasmid transfection**. Cells were plated at 200,000 cells/well in a 6-well plate, or 100,000 cells/well in a 12-well plate, the day prior to transfection. Each cell line was transfected with increasing quantity of TMOs with two different transfection agents (Lipofectamine RNAiMAX (Thermo Fisher Scientific) and Xtreme Gene siRNA transfection agent (Sigma)) to determine the optimal transfection conditions for each cell line. Fluorescently labeled TMO was used to assess transfection efficiency, while intron-spanning RT-PCR (only for TUG1), RT-qPCR and smRNA FISH were used to assess the efficiency of intron inclusion. Lipid-oligo complexes were prepared at room temperature in OptiMem medium (Thermo Fisher Scientific) according to the manufacturer's instructions. After incubation time, lipid-oligo complexes were added dropwise to wells containing freshly added full growth media. U-2 OS was most efficiently transfected with Xtreme Gene, while HeLa, LN-18 and HEK293T cell lines were more efficiently transfected with Lipofectamine RNAiMAX. 25 nM TMO was chosen as the lowest quantity achieving maximum intron inclusion efficiency. 1 μg of a plasmid expressing hTERT (kind gift from Joachim Lingner, École polytechnique fédérale de Lausanne (EPFL), Lausanne, Switzerland[78]) was transiently co-transfected with corresponding TMO as described above.

**Cell viability assays**. Cells were plated at density of 1000 cells/well in a 96 well plate. After 24 h, cells were transfected with 25 nM of the corresponding TMO. 48

h and 72 h post-transfection, cell culture media was replaced by 10% of AlamarBlue reagent (DAL1100, ThermoFisher Scientific) in full growth media 2–4 h prior to reading fluorescence. Fluorescent data was collected using the CLARIOstar microplate reader from BMG Labtech fluorescence plate reader following the manufacturer's recommendations.

**Cell cycle analysis with flow cytometry**. Cells were washed with PBS, collected with trypsinization and spun down. After a rinse with PBS, cells were fixed in 70% ethanol for at least 2 h at −20 °C. Cells were spun down; supernatant was removed and cells were resuspended in a PBS containing 35 μg/mL propidium iodide (Sigma) and 100 μg/mL RNAse A (Roche). Data was acquired on a BD FACS-Celesta Flow Cytometer and analyzed with ModFit LT 5.0 software. Gating strategy used to identify single-cell populations is shown in Supplementary Fig. 12.

**DNA extraction**. Genomic DNA was extracted from pelleted cells using Monarch® Genomic DNA Purification Kit (NEB).

**Southern blot**. Telomere length analysis via Southern blotting was performed as described previously[40]. Briefly, ~1.5 μg of genomic DNA from each cell sample was digested with RsaI and HinfI. Fragments were resolved on a 0.8% agarose gel, and then transferred to Hybond N + Nylon membrane (GE) via capillary transfer. The fragments were crosslinked to the membrane, and the membrane was probed for telomeric sequence using radiolabeled (TTAGGG)₄ DNA oligo. After washing off nonspecifically bound probe, the membrane was exposed on a phosphor screen and imaged using a Typhoon FLA 9500 Variable Mode Imager (GE). Median telomere length was calculated by extracting signal profiles from each lane using ImageJ, finding the median point of each distribution using Microsoft Excel, and converting those pixel coordinates into base pair lengths using a λ-HindIII molecular weight marker (NEB).

**TUG1 and TERT conservation analysis**. Human and mouse TUG1 and TERT genomic sequences were downloaded from hg38 and mm10, respectively. Alignments were prepared in Geneious using MAFFT v7.388[79,80]. Alignments were imported in CLC main workbench (Qiagen) where sequence conservation was further analyzed by pairwise sequence comparison and visualized.

**Splice site strength analysis**. MaxEntScan[81] was used to calculate maximum entropy scores for 9 nt donor splice sites and 20 nt acceptor splice sites.

**RNA sequencing and read alignment**. Total RNA from U-2 OS, HeLa and mES cell lines was extracted with Maxwell LEV Simply RNA isolation kit. RNA quality was assessed with BioAnalyzer. One microgram of total RNA from U-2 OS was subjected to poly(A) RNA enrichment and library preparation with NEBNext® Poly(A) mRNA Magnetic Isolation Module (NEB E7490) and NEBNext® Ultra RNA Library Prep Kit from Illumina® (E7530), and sequenced on the HiSeq4000. One microgram of total RNA from HeLa and mES was subjected to poly(A) RNA enrichment and library preparation with TruSeq kit v2 according to manufacturer's instructions and sequenced on HiSeq4000 (HeLa) or HiSeq2500 (mES). We retrieved RNA-seq data for HEK293 (poly(A)+), LN-18 (poly(A)+), HCT116 (poly (A)+) and fibroblasts (poly(A)+) (accession numbers: HEK293: SRR3997506, LN-18: SRR8769945, HCT116: SRR8615282, fibroblasts: SRR5420980) and gene annotations were retrieved from Gencode (vM23). Raw reads were mapped to GRCm38 using the nf-core RNA-seq pipeline (v1.4.2)[82]. Raw reads from poly(A)+ RNA sequencing of chromatin-associated, nuclear soluble, and cytoplasmic fractions of mES cells were retrieved (GSE80262)[83]. Relative abundance in each fraction was calculated from transcript per million TPM values (for instance, chromatin TPM divided by the sum of TPMs in all compartments).

**Primers and TMOs**. List of primer and TMO sequences used in this study are listed in Supplementary Table 1.

**Reporting summary**. Further information on research design is available in the Nature Research Reporting Summary linked to this article.

## Data availability
The data supporting the findings of this study are available from the corresponding authors upon reasonable request. The sequencing data are accessible through GEO Series accession number GSE169743. Browser tracks can be found at: https://genome.ucsc.edu/s/GabrijelaD/TERT_multiple_cell_lines_Share (HEK293, LN-18, HCT116, HeLa and BJ fibroblasts for TERT); https://genome.ucsc.edu/s/GabrijelaD/TUG1_multiple_cell_lines_Share (U2-OS, HeLa, BJ fibroblasts, LN-18, HEK292, HCT116 for TUG1). Source data are provided with this paper.

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

# ARTICLE

61. Li, Y., Zhang, T., Zhang, Y. & Wang, W. Targeting the FOXM1- regulated long noncoding RNA TUG1 in osteosarcoma. *Cancer Sci.* **109**, 3093–3104 (2018).

62. Nakashima, M., Nandakumar, J., Sullivan, K. D., Espinosa, J. M. & Cech, T. R. Inhibition of telomerase recruitment and cancer cell death. *JCB* **288**, 33171–33180 (2013).

63. Li, S., Crothers, J., Haqq, C. M. & Blackburn, E. H. Cellular and gene expression responses involved in the rapid growth inhibition of human cancer cells by RNA interference-mediated depletion of telomerase RNA. *J. Biol. Chem.* **280**, 23709–23717 (2005).

64. Fatemi, A., Safa, M. & Kazemi, A. MST-312 induces G2/M cell cycle arrest and apoptosis in APL cells through inhibition of telomerase activity and suppression of NF-κB pathway. *Tumor Biol.* **36**, 8425–8437 (2015).

65. Celeghin, A. et al. Short-term inhibition of TERT induces telomere length-independent cell cycle arrest and apoptotic response in EBV-immortalized and transformed B cells. *Cell Death Dis.* **7**, 1–11 (2016).

66. Thompson, C. A. H. et al. Transient telomerase inhibition with imetelstat impacts DNA damage signals and cell-cycle kinetics. *Mol. Cancer Res.* **16**, 1215–1225 (2018).

67. Lubelsky, Y. & Ulitsky, I. Sequences enriched in Alu repeats drive nuclear localization of long RNAs in human cells. *Nature* **555**, 107–111 (2018).

68. Shukla, C. J. et al. High-throughput identification of RNA nuclear enrichment sequences. *EMBO J.* **37**, e98452 (2018).

69. Wright, W. E., Tesmer, V. M., Liao, M. L. & Shay, J. W. Normal human telomeres are not late replicating. *Exp. Cell Res.* **499**, 492–499 (1999).

70. Zou, Y., Gryaznov, S. M., Shay, J. W., Wright, W. E. & Cornforth, M. N. Asynchronous replication timing of telomeres at opposite arms of mammalian chromosomes. *Proc. Natl Acad. Sci. USA* **101**, 12928–12933 (2004).

71. Arnoult, N. et al. Replication timing of human telomeres Is chromosome arm–specific, influenced by subtelomeric structures and connected to nuclear localization. *PLoS Genet.* **6**, e1000920 (2010).

72. Dominguez, D. et al. An extensive program of periodic alternative splicing linked to cell cycle progression. *Elife* **5**, e10288 (2016).

73. Golipour, A. et al. A late transition in somatic cell reprogramming requires regulators distinct from the pluripotency network. *Cell Stem Cell* **11**, 769–782 (2012).

74. Gu, Z., Eils, R. & Schlesner, M. Complex heatmaps reveal patterns and correlations in multidimensional genomic data. *Bioinformatics* **32**, 2847–2849 (2016).

75. Bailey, T. L. et al. MEME Suite: tools for motif discovery and searching. *Nucleic Acids Res.* **37**, 202–208 (2009).

76. Giudice, G., Sánchez-Cabo, F., Torroja, C. & Lara-Pezzi, E. ATtRACT-a database of RNA-binding proteins and associated motifs. *Database* 1–9 https://doi.org/10.1093/database/baw035 (2016).

77. Krotz, A. H., Mcelroy, B., Scozzari, A. N., Cole, D. L. & Ravikumar, V. T. Controlled deacetylation of antisense oligonucleotides. *Org. Process Res. Dev.* **7**, 47–52 (2003).

78. Cristofari, G. & Lingner, J. Telomere length homeostasis requires that telomerase levels are limiting. *EMBO J.* **25**, 565–574 (2006).

79. Katoh, K., Misawa, K., Kuma, K. & Miyata, T. MAFFT: a novel method for rapid multiple sequence alignment based on fast Fourier transform. *Nucleic Acids Res.* **30**, 3059–3066 (2002).

80. Katoh, K. & Standley, D. M. MAFFT multiple sequence alignment software version 7: improvements in performance and usability. *Mol. Biol. Evol.* **30**, 772–780 (2013).

81. Yeo, G. & Burge, C. Maximum entropy modeling of short sequence motifs with applications to RNA splicing signals. *J. Comput. Biol.* **11**, 377–394 (2004).

82. Ewels, P. A. et al. The nf-core framework for community-curated bioinformatics pipelines. *Nat. Biotechnol.* **38**, 271–278 (2020).

83. Engreitz, J. M. et al. Local regulation of gene expression by lncRNA promoters, transcription and splicing. *Nature* **539**, 452–455 (2016).

## Acknowledgements

We thank the Rinn lab members for insightful discussions. We thank Roy Parker and Carolyn Decker (University of Colorado Boulder) for access to the DeltaVision Elite microscope. We thank Theresa Nahreini and Nicole Kethley for use of the Cell Culture Facility (University of Colorado Boulder). iPS and LN-18 cells were a gift from Teisha Rowland from the Stem Cell Research and Technology Resource Center (University of Colorado Boulder). We thank Arthur Zaug (T.R.C. laboratory) for assistance with the TERT overexpression and Southern blot experiments. We thank Manuel Irimia (CRG, Barcelona) for sharing vast-tools results from various tissue/cell types from seven mammalian species. U.B. is supported by funds from a Canadian Institutes for Health Research Foundation Grant to B.B. T.R.C. is an investigator of the Howard Hughes Medical Institute (HHMI). J.L.R. is an HHMI Faculty Scholar and holds a Marvin H. Caruthers Endowed Chair for Early Career Faculty. This research was supported by NIH grant P01GM099117 to J.L.R.

## Author contributions

Study conceptualization and design: G.D., T.R.C., M.C., and J.L.R.; experiment design, performance, microscopy and data analysis: G.D.; Vast-tools analysis: U.B.; TMO synthesis: H.K. and K.J.; computational analyses: G.D., U.B., and M.S.; cell cycle analysis: J.B.; Southern blot: E.H., intellectual input: G.D., U.B., H.K., J.B., B.B., T.R.C., M.C. and J.L.R.; funding: M.C. and J.L.R.; writing the paper: G.D. and J.L.R. with input from all of the authors.

## Competing interests

The authors declare no competing interests. T.R.C. is on the Merck board and is a consultant for Storm Therapeutics and Eikon Therapeutics.

## Additional information



