## [Peer Review File · Nature Communications]

REVIEWER COMMENTS

Reviewer #1 (Remarks to the Author):

In the manuscript "Nuclear compartmentalization of TERT mRNA and TUG1 lncRNA transcripts is driven by intron retention: implications for RNA-directed therapies", Dumbović et al. examine subcellular localization of TERT mRNA and TUG1 lncRNA using smRNA FISH in various human cell lines and mouse embryonic stem cells. Using probe-sets targeting exons and introns, they characterize splicing efficiency and nuclear enrichment of spliced and unspliced transcripts and identify stable, nuclear-enriched unspliced transcripts in steady state cells. Splicing and localization dynamics were also tested in different stages of the cell cycle, and the authors identify mitosis-specific splicing event of one of the introns that are typically retained in the TERT mRNA. The authors examine the conservation of splicing and localization of TERT and TUG1 between human and mouse using bioinformatic analysis and smRNA FISH in mouse ESCs. Finally, the authors use thiomorpholino antisense oligonucleotides (TMOs) to inhibit splicing and show that this treatment efficiently blocks intron excision and results in strong nuclear restriction of both TERT and TUG1 transcripts and in disruption of cell growth.

Overall, the manuscript is well written and the association between intron retention and nuclear restriction is well supported by the examined experimental system. How RNA localization is regulated is a topic of broad interest, and there is accumulating evidence over several decades of inter-connections between splicing and export. The link between intron retention and nuclear compartmentalization was demonstrated by several studies that examined other specific transcripts, reporters and global transcriptome profiling to date. This manuscript adds two detailed and well supported examples for this connection and provides evidence that processing efficacy, inducing a specific cytoplasmic/nuclear ratio, may be functionally important in these two genes.

Major comments

1. Regarding the two genes, several mechanistic questions remain open: Why splicing of some specific introns is consistently inefficient? What cellular machinery and/or intronic sequence is responsible for retaining the unspliced transcripts in the nucleus? The authors did not address these questions, besides Extended data Fig. 5, which provides very limited insight into the mechanism. A deeper bioinformatic analysis can potentially shed some light on some of these questions, either by looking for known sequence motifs or RNA binding protein (RBP) recognition sites in these introns, or by examining available CLIP datasets to identify binding of RBPs specifically to these retained introns.
2. One of the important questions the authors address in this study is the conservation of selective intron retention (Extended data figure 4). This analysis can be expanded and moved to the front as a main figure, and can contribute to the novelty of the results. Generally, the authors considered only conservation between human and mouse. This comparison will be clearer by showing human and mouse quantification side by side (as in Extended data figure 4b, add the human cell lines) and could be complemented by examining splicing of TERT and TUG1 in some other species using available RNA-seq datasets in a similar manner. Finally, the authors found that distinct introns are retained in human and mouse. This is an interesting finding, and can be examined from mechanistic point of view as discussed above. For example, are specific sequence motifs shared between the introns that are retained in human and mouse?
3. The authors use the TMO treatment as evidence for the functional importance of a particular splicing state that dictates a particular localization pattern, as the TMO treatment affects splicing and localization, and also causes a proliferation defect. A concern with this experiment is that the TMOs themselves might be toxic, through off-target effects, or through general toxicity that is often observed with these kinds of reagents. A pretty straightforward way to address this would be to test what happens when cells are first depleted of TUG1 or TERT (e.g., using GapmeRs that should be quite effective in depleting nuclear transcripts) and then treated with the TMOs. In this case it is expected that the TMOs will not have an effect on proliferation beyond that observed when the

transcripts are depleted.

Minor points

1. The authors mention "implications for RNA-directed therapies", but this point is not really addressed in any specific experiments in the manuscript, so is probably not fit for the title.
2. In several figures in this study, the authors use scatter plots showing nuclear exon vs. nuclear intron counts. As these plots are meant to show the correlation between intron retention and nuclear enrichment, it will be more appropriate to convert all these plots to show total PIR vs percent of nuclear retention, as in Fig. 2C for the different cell lines.
3. Typo in line 27 – "thiomorpholino"
4. Fig. 1C and line 101 – "TUG1 intron 2 is absent from the VastDB human database" – it is probably possible to compute PIR for intron 2, either by manual annotation input for VastDB or using any other tool for splicing quantification that can be provided with a custom annotation file.
5. Fig. 1F - The cells in the images are so dense, that it is not possible to see the cytoplasm, how it is possible to quantify cytoplasmic signal? Also, it is not clear from the Microscopy and image analysis section (in Methods) how cell segmentation was performed for further cytoplasmic quantification.
6. Extended data Fig. 4a –
 - o It's misleading to show these two genes side by side, since TERT is mostly long intronic sequence, not well conserved as expected, while TUG1 has relatively short introns. Maybe show just CDS of TERT, and then the relevant introns separately.
 - o This should be shown aligned to an RNA-seq track, to make it possible to assess the conservation of the retained introns.
 - o I'm not sure what the color gradient on the right means, if conservation here is per nucleotide it is not a continuous rather a binary variable. Can be removed or better explained.
7. Fig. 3C (right) should be presented same as Fig. 2C (right), this is more clear and informative. In addition, since both Fig. 2 and Fig. 3 are very similar and convey the same message, one of them can be moved to supplement.
8. Extended data Fig. 5 - Splitting this figure to panels and adding references throughout the text will make the corresponding paragraph easier to follow
9. Fig. 4 – This analysis does not provide substantial new insights, as splicing differences between coding genes and lncRNAs are known. It can be moved to the supplemental material.
10. Line 288 – it seems that a few sentences were dropped here, since cell cycle was not mentioned up to this point? How was the cell cycle phase determined? In how many cells? Etc.
11. In Fig. 7E, it seems clear that some alternative splicing is happening in both TUG1 introns. For intron 1, the TMO treatment clearly affects one of these alternative isoforms. In current RefSeq annotations these events are possibly represented by multiple isoforms. It is interesting to test using available RNA-seq data and/or RT-PCR the identity of these alternative isoforms, their subcellular localization, and the effect of TMOs on their efficiency.
12. Line 341 – "Fig. 7F" (typo).
13. In Fig 7C you use "Cell growth assays", while in Fig. 8B you use "Cell line viability assay" – are these different assays?

Reviewer #2 (Remarks to the Author):

Dumbovic et al. study the nuclear retention of TERT mRNA and TUG1 lncRNA and show that intron retention is involved. Conservation of this mechanism between humans and mice is shown. They also develop a method to perturb intron excision that could have therapeutic use. I would recommend dividing this into two papers as the two RNAs studied don't seem to have a lot in common. If the authors don't want to do this, they need to at least expand discussion of TUG1 RNA to make it interesting. Further, they could extend this analysis by investigating the subnuclear location of the retained RNAs.

1. Nuclear retention of TERT mRNA by specific intron retention and its further splicing and export during mitosis is a very exciting and important finding. This could explain its activation in cancer cells. Further analysis of its subnuclear localization would be very interesting—is it in speckles? Chromatin?. It is obviously an important translated mRNA so nuclear retention must be a regulatory mechanism.
2. TUG1 lncRNA is not described in much detail as to its normal function and localization. It is said to have a role in many cell processes and to be oncogenic. It doesn't seem unusual for an unspliced lncRNA to be nuclear and perhaps spliced and exported for degradation. Where in the nucleus is it located? Is it chromatin-associated? It's not clear what it has in common with TERT except both are up-regulated in cancer cells and have retained introns.
3. Line 99 (and 215) say that TERT intron 11 and 14 PIR are 30 and 31%, which made me wonder how that caused most RNA to be nuclear. Are intron 11 and 14 both retained on the same RNA or different RNAs? However, Line 122 says TERT intron 11 has a nuclear PIR of 90%. Please explain
4. It appears that TERT intron 2 is also retained in some cell lines but is downplayed here. It seems possible that skipping exon 2 (and adjacent introns) would minimize its retention. Line 107: It seems like a poor control for a non-retained intron. What was the PIR for GAPDH intron 2?
5. Page 5: what does it mean that TUG1 intron 2 is absent from the VastDB human database? Is it not retained?
6. How much mouse TERT mRNA is retained in the nucleus? I couldn't find this info.
7. It is interesting that many other mRNAs are retained in the nucleus. Are they cell cycle regulated also?
8. Extended data Table 3b: Some bedgraphs appear to be missing.
9. Line 104: Tiling not tiling

Reviewer #3 (Remarks to the Author):

In this manuscript, Dumbovic et al. compare the nuclear/cytoplasmic distribution of fully spliced RNAs to their intron-retained counterparts. They use smRNA FISH as their primary technique to investigate localization of TERT mRNA and TUG1 ncRNA. Overall, the data are clear, believable, and well presented in a nicely focused paper. The writing is generally good, and I do not have any major criticisms of the work that is presented. The work is descriptive and none of the observations are particularly surprising. In fact, unspliced introns leading to nuclear retention has been explored since the late 80s with the pioneering work from Sharp and Rosbash labs and recent global analyses suggest that intron retention is strongly associated with nuclear retention in mammals. Therefore, one may quibble about whether this work warrants publication in non-specialist journal like Nature Communications. Nonetheless, I would say the strength of the scientific approach and the clarity of the data support its publication there.

I do have a few minor, but important, suggestions for improvement:

1. In Fig 1 and throughout. Please use the figure legends and more intuitive axis labels to clarify what the graphs are. Just a few examples: What is "signal count" (same as "spots" used later)? Does "FISH signal" mean intensity or number of spots and is it the same as "signal count"? I have no idea what is being plotted in the middle graphs on Fig 1e and 1f. The grey and purple make for nice depiction of

data and I pretty sure that I understand the point, but I have no idea what is being plotted. The X-axis is labeled "nucleus"; what does that mean? Does an N=50 mean number of cells or spots counted? If the latter, it seems like a small number of cells to count for this kind of analysis without demonstrating that these are representative cells. How many independent replicates of the experiment was performed?

2. In the RNA-seq data and RT-qPCR data, please state in the Results or Figure legends whether or not the RNAs were primed by dT or random priming. This makes a big difference in the interpretation of whether these are fully processed RNAs with retained introns or whether they are potentially RNAs being transcribed and not yet processed.

3. One should be cautious regarding the ActD experiment interpretation. Nuclear RNAs with retained introns are hyperadenylated and may be artificially stabilized upon transcription inhibition (e.g. Bresson et al. 2015).

4. Line 288 is confusing—what is being referred to by "the above smRNA FISH analyses revealed that TERT pre-mRNA was spliced during cell division"?

5. The interpretations of the TERT/mitosis data are too strong. There is a robust correlation and their data are believable, but without orthogonal approaches, it is too strong to say that "nuclear TERT transcripts uniformly and robustly retain two specific introns whose splicing occurs during mitosis". Their data form a solid basis for this hypothesis, but it's far from fully proven. It's fine to include the data, but softening the interpretations is necessary.

6. Line 341: 7f not 6f

7. The statement on detained introns (line 432) implies that detained introns are defined as those that serve as reservoirs for posttranscriptional splicing. While some certainly do (e.g. Mauger et al. 2016), the definition of "detained" by Boutz et al. (2015), is simply that these are transcripts that have retained introns that are not exported to the cytoplasm. In principle, such RNAs could be reservoirs, nonfunctional partially spliced byproducts, or serve as nuclear noncoding RNAs.

We thank the reviewers for their positive assessment of our manuscript “**Nuclear compartmentalization of TERT mRNA and TUG1 lncRNA is driven by intron retention**”. Their thorough comments, detailed and insightful suggestions were helpful to better contextualize the results and to provide additional analyses which strengthen the manuscript. In our revised work we addressed comments from all reviewers. We edited the manuscript and figures according to their suggestions, and denoted all changes in the manuscript in blue to aid the reviewer’s assessment. Please find below a point-by-point response to each reviewers' comment.

REVIEWER COMMENTS

Reviewer #1 (Remarks to the Author):

In the manuscript “Nuclear compartmentalization of TERT mRNA and TUG1 lncRNA transcripts is driven by intron retention: implications for RNA-directed therapies”, Dumbović et al. examine subcellular localization of TERT mRNA and TUG1 lncRNA using smRNA FISH in various human cell lines and mouse embryonic stem cells. Using probe-sets targeting exons and introns, they characterize splicing efficiency and nuclear enrichment of spliced and unspliced transcripts and identify stable, nuclear-enriched unspliced transcripts in steady state cells. Splicing and localization dynamics were also tested in different stages of the cell cycle, and the authors identify mitosis-specific splicing event of one of the introns that are typically retained in the TERT mRNA. The authors examine the conservation of splicing and localization of TERT and TUG1 between human and mouse using bioinformatic analysis and smRNA FISH in mouse ESCs. Finally, the authors use thiomorpholino antisense oligonucleotides (TMOs) to inhibit splicing and show that this treatment efficiently blocks intron excision and results in strong nuclear restriction of both TERT and TUG1 transcripts and in disruption of cell growth.

Overall, the manuscript is well written and the association between intron retention and nuclear restriction is well supported by the examined experimental system. How RNA localization is regulated is a topic of broad interest, and there is accumulating evidence over several decades of inter-connections between splicing and export. The link between intron retention and nuclear compartmentalization was demonstrated by several studies that examined other specific transcripts, reporters and global transcriptome profiling to date. This manuscript adds two detailed and well supported examples for this connection and provides evidence that processing efficacy, inducing a specific cytoplasmic/nuclear ratio, may be functionally important in these two genes.

We thank the reviewer for the positive assessment of our manuscript and their suggestions.

Major comments

R1.1. Regarding the two genes, several mechanistic questions remain open: Why splicing of some specific introns is consistently inefficient? What cellular machinery and/or intronic sequence is responsible for retaining the unspliced transcripts in the nucleus? The authors did not address these questions, besides Extended data Fig. 5, which provides very limited insight into the mechanism. A deeper bioinformatic analysis can potentially shed some light on some of these questions, either by looking for known sequence motifs or RNA binding protein (RBP) recognition sites in these introns, or by examining available CLIP datasets to identify binding of RBPs specifically to these retained introns.

The reviewer raises several interesting and important questions. The question of what determines retention of introns in general has been the focus of multiple studies ranging from single examples to transcriptome-wide (e.g., Braunschweig et al., *Genome Res.* 2014; Boutz et al., *Genes Dev.* 2015; Wong JJ et al., *Nature Comms.* 2017; Yeom et al., *bioRxiv* doi:<https://doi.org/10.1101/2020.10.23.352088>). However, a consistent model has so far been hard to elucidate, and the regulation of a particular intron is still hard to predict from sequence.

Based on the reviewer's suggestion we wanted to shed some light on this question in the context of the introns retained in human iPS cells. We first took a general approach and overlapped retained introns with eCLIP of 103 RNA-binding proteins in HepG2 and 120 in K562 cells, obtained by the Yeo lab (van Nostrand et al., Nature 2020). Indeed, we found several RBPs enriched over these introns in both HepG2 and K562, including PRPF8, U2AF1, U2AF2, AQR, EFTUD2, QKI, RBFOX2, HNRNPK and PTBP1. We further analyzed RNA-seq data from the ENCODE consortium in which these factors were depleted, and found that several were significantly enriched in introns whose retention increases (PTBP1) or decreases (U2AF1, U2AF2, PRPF8) upon depletion. However, the interpretation of these results is ambiguous (Are they enriched because retained introns are hard to recognize by the spliceosome resulting in longer dwell times? Or does their binding promote retention of certain introns?), and we feel that inclusion of these results would not strengthen the manuscript.

To assess the influence of RNA binding proteins directly on TUG1/TERT intron retention we took a multifaceted approach and analyzed RBP motif instances (retrieved from the AtTRACT database) and binding events detected across all of ENCODE's eCLIP datasets in TUG1 and TERT introns. We visualized peak coverage across introns as well as motif instances. No clear candidates stood out as uniquely binding to retained introns or excluded from retained introns. To aid in interpreting the results, a random forest regression was performed on genome-wide intron retention levels, attempting to learn the relationship between RBP binding (represented by eCLIP peak coverage in each intron) and intron retention. The top 20 RBPs most useful in predicting intron retention levels (determined by random forest variable importance) were highlighted and the direction of influence was determined by a linear regression model of the same binding vectors. Both RBPs predicted to increase splicing efficiency (PRPF8, AQR for ex.) and RBPs predicted to increase retention (YBX3, AKAP1 for ex.) are found to be strongly bound across both TUG1 introns – therefore precluding any definite conclusions from this analysis (Reviewer Fig. 1). Regarding TERT intron binding patterns, we observed that RBP binding seems enriched overall in the retained introns, but whether this is due to intron retention (therefore increasing detectability in eCLIP) or is driving intron retention is unknown. We added a section and a figure to the paper relating these results (lines 272-284, Extended data Fig. 5 and Extended data Fig. 6).

Lastly, to analyze the functions of the RBPs identified by eCLIP in TERT/TUG1 intron retention, we further analyzed RNA-seq data from the ENCODE consortium in which these factors were depleted. The analysis was met with certain caveats. TERT expression is too low in both HepG2 and K562 to measure retention changes for most introns; and TUG1 intron retention changed moderately in several of the RNAi datasets (for instance, PTBP1). To experimentally validate the involvement of PTBP1 in TUG1 intron retention, we analyzed changes in TUG1 intron retention by smRNA FISH after knocking down PTBP1. We did not detect notable changes, thus a direct involvement of the depleted RBP remains elusive. We have therefore decided against inclusion of these results in the revised manuscript and respectfully suggest that directly addressing these mechanistic questions with new experiments would be beyond the scope of this manuscript.

Reviewer Fig. 1. TUG1 intronic RNA binding proteins and motifs. a, Heatmaps of the eCLIP fractional peak coverage within each intron window (40bp up and downstream from the 5' and 3' splice site & five partitioned windows in the intron interior). A value of 1 indicates that all base pairs of that window were covered by an eCLIP peak. The heatmap to the right highlights those RBPs which were predictive of intron retention genome-wide and the color represents the direction of influence (red indicates an increase in the retention level if that protein was bound, blue indicates a predicted decrease). **b**, The top 20 most predictive RBPs determined by random forest variable importance. The y-axis shows the linear model regression coefficients to give a sense of the likely influence on retention (positive would indicate a likelihood to increase retention). **c**, Motif instance matches in TUG1. In each window, the color represents the maximum scoring motif match in the window as calculated by FIMO (motifs only shown with $p < 1e-4$).

a TERT eCLIP Peak Coverage

Extended data figure 5: eCLIP and RBPs motifs across TERT introns. a, Heatmap colored by total intron covered by eCLIP peaks for each of the 127 RBPs with TERT intron binding. **b,** Motif instance matches in TERT introns. In each window, the color represents the maximum scoring motif match in the window as calculated by FIMO (motifs only shown with $p < 1e-4$).

b

Response to Reviewers Nature communications NCOMMS-20-27782A

Extended data figure 6: RNA binding protein (RBP) occupancy is increased over retained TERT introns. a, Total fraction of each intron window covered by eCLIP peaks. The 15 blocks of data represent the 15 TERT introns. Each block is then divided into windows, which are 40bp upstream and downstream of the 5' and 3' splice sites; intron interiors are partitioned into five windows. **b,** Heatmap colored by fraction of window covered by eCLIP peaks for each of the 127 RBPs with TERT intron binding.

R1.2. One of the important questions the authors address in this study is the conservation of selective intron retention (Extended data figure 4). This analysis can be expanded and moved to the front as a main figure, and can contribute to the novelty of the results. Generally, the authors considered only conservation between human and mouse. This comparison will be clearer by showing human and mouse quantification side by side (as in Extended data figure 4b, add the human cell lines) and could be complemented by examining splicing of TERT and TUG1 in some other species using available RNA-seq datasets in a similar manner. Finally, the authors found that distinct introns are retained in human and mouse. This is an interesting finding, and can be examined from mechanistic point of view as discussed above. For example, are specific sequence motifs shared between the introns that are retained in human and mouse?

The reviewer makes an important point and suggestion.

Accordingly, we have now moved the Extended data figure 4 to Figure 4. We extended the evolutionary analysis to Vast-tools results for introns in TERT in seven mammalian species across multiple tissue and cell line samples (*H. sapiens*, 128; *P. troglodytes*, 38; *M. mulatta*, 66; *R. norvegicus*, 127; *M. musculus*, 151; *B. taurus*, 58; *M. domestica*, 44). TUG1 splice sites are absent from the VastDB database from most species where the TUG1 gene is not annotated, therefore we focused on TERT. The analysis revealed very intriguing difference in selective intron retention, even among mammals of the same order. The retention of intron 11 is conserved among primates. Retention of intron 14 appears to be primarily a human-specific phenomenon. Variability exists in retention of other introns (for instance, intron 15 is highly retained in macaque). High retention of introns 3 and 7 is conserved between rat and mouse, while rat Tert additionally retains other introns, including intron 11. These data demonstrate that both intron sequences and intron retention have undergone evolutionary divergence between mammalian orders. Intriguingly, TERT contains retained introns in all species investigated, thus opening the possibility that regulation of TERT via subcellular localization may be an evolutionarily conserved mechanism. We have added the new data to Figure 4, which has been additionally modified to accommodate all the reviewer's suggestions (including minor point 6), and the corresponding text (lines 231-242).

R1.3. The authors use the TMO treatment as evidence for the functional importance of a particular splicing state that dictates a particular localization pattern, as the TMO treatment affects splicing and localization, and also causes a proliferation defect. A concern with this experiment is that the TMOs themselves might be toxic, through off-target effects, or through general toxicity that is often observed with these kinds of reagents. A pretty straightforward way to address this would be to test what happens when cells are first depleted of TUG1 or TERT (e.g., using GapmeRs that should be quite effective in depleting nuclear transcripts) and then treated with the TMOs. In this case it is expected that the TMOs will not have an effect on proliferation beyond that observed when the transcripts are depleted.

We agree that excluding potential off targets would further support the specificity of this approach. Depleting the RNA before adding the TMOs is a good strategy, however, not feasible in our assays since the cells would have to be transfected two consecutive times, which would affect cell viability and prolong the treatment, during which the control cells would reach confluency. Additionally, downregulating first the RNAs of important genes such as *TERT* and *TUG1* could affect the proliferation of the cells, making the cell viability analysis less conclusive.

However, we wanted to address the reviewers point. To this end, we performed a rescue experiment with a fully spliced TERT mRNA, transfected at the same time as the TERT TMO. The fully spliced TERT mRNA is not a target of TERT TMO, hence it will be able to compensate for the lack of endogenous spliced TERT. First, the analysis of the cell cycle in LN-18 and HEK293T cells that co-express TERT and TUG1 show that TERT TMO causes a specific arrest in G2/M phase and induction of γ H2A.X foci compared to control and TUG1 TMOs (Extended data Fig. 10b,c). This is in accordance with previous studies which show that short term inhibition of telomerase leads to an induction of γ H2A.X and cell cycle arrest through telomere length-independent processes.

We next sought to determine whether the specific proliferation, cell cycle and genomic damage phenotype caused by inhibiting TERT splicing with TERT TMO would be absent if we co-express fully spliced TERT. Indeed, we observed that the proliferation, cell cycle and genomic damage are reduced to the levels comparable to control TMO once spliced TERT is present (Fig. 8f). These results support the notion that the observed cell viability changes are TERT specific. These results have been added to Figure 8f and Extended data Fig. 10, and the corresponding text has been added to lines 460-469.

Extended data Fig. 10: a, Telomere restriction fragment (TRF) assay of LN-18 cells treated with transfection agent only, control TMO or TERT TMO up to 120 h. Sizes are shown in base pairs and dot plot indicates the mean telomere size for each lane. **b**, Cell viability and cell cycle analysis of HEK293T and LN-18 cells 72 h after transfection with TUG1 TMOs, TERT TMO or control TMO compared to control TMO. Error bars represent SD, $n =$ three independent measurements. **c**, Quantification of γ H2A.X foci of experimental conditions shown in b. Representative images of γ H2A.X immunofluorescence are shown, DAPI, blue; γ H2A.X, green; scale bar 5 μ m. HEK293T, $n =$ 50-52 cells; LN-18, $n =$ 50-51 cells.

Figure 8f. Cell viability, cell cycle and γ H2A.X foci analysis of LN-18 cell line 72 h after co-transfection of TERT TMO with spliced TERT compared to control TMO or TERT TMO co-transfected with a control plasmid. Cell viability and cell cycle; error bars represent SD, $n = 3$ independent measurements. Representative images of γ H2A.X immunofluorescence are shown, DAPI, blue; γ H2A.X, green; scale bar 5 μ m. Violin plot, $n = 129$ -169 cells, two independent stainings.

Minor points

1. The authors mention “implications for RNA-directed therapies”, but this point is not really addressed in any specific experiments in the manuscript, so is probably not fit for the title.

We agree with the reviewer and have changed the title accordingly.

2. In several figures in this study, the authors use scatter plots showing nuclear exon vs. nuclear intron counts. As these plots are meant to show the correlation between intron retention and nuclear enrichment, it will be more appropriate to convert all these plots to show total PIR vs percent of nuclear retention, as in Fig. 2C for the different cell lines.

With these plots, we wanted to show that the extent of intron retention in nuclear transcripts is high and consistent between individual cells (or not, in case of TERT intron 2 and intron 14). While we do see a correlation between total PIR and percent of nuclear retention for TUG1 intron 1 and intron 2 between individual cells, this correlation does not exist for TERT. This is mainly because regardless of the cell type or individual cell, the number of molecules is low and the nuclear/cytoplasmic ratio consistent (Fig. 3b,c). On the other hand, TUG1 nuclear enrichment varies between cell lines (Extended data Fig. 2b). Thus, with Fig. 2c we aimed to show that there is a correlation between total PIR and nuclear enrichment for TUG1 between cell types.

3. Typo in line 27 – “thiomorpholino”

We corrected the typo.

4. Fig. 1C and line 101 – “TUG1 intron 2 is absent from the VastDB human database” – it is probably possible to compute PIR for intron 2, either by manual annotation input for VastDB or using any other tool for splicing quantification that can be provided with a custom annotation file.

Unfortunately, vast-tools (which underlies VastDB) is not amenable to easy alterations to its database, which makes it infeasible to include splice junctions. While it would be technically possible to map RNA-seq reads to a set of junctions representing the missing intron, this would not make the resulting measurements comparable with vast-tools results. Instead, we refer the reviewer to our smRNA FISH results (Fig. 1 and 2), which we believe provide a clear readout for this intron relative to the other studied introns.

5. Fig. 1F - The cells in the images are so dense, that it is not possible to see the cytoplasm, how it is possible to quantify cytoplasmic signal? Also, it is not clear from the Microscopy and image analysis section (in Methods) how cell segmentation was performed for further cytoplasmic quantification.

For an accurate manual cell segmentation, cell borders were visualized in ImageJ upon increasing the saturation of the FISH channel. For all cell types the cell border could be readily visualized, with the exception of iPS cells. They grow at higher confluency; hence it is more challenging to see cytoplasm contours upon confluency. To overcome this, we plated the cells the day before RNA FISH to ensure the colonies were still at lower density, and only non-overlapping cells where cytoplasm borders could be visualized were quantified (Reviewer Fig. 2). We added a more detailed explanation of the quantification process to the Methods section under Single molecule RNA FISH and Microscopy and Image analysis:

Lines 624-625: *“The iPS cells were seeded at lower density the day before harvesting the coverslips to facilitate the quantification process.”*

Lines 659-661: *“To outline cell borders, we used manual segmentation. For an accurate manual cell segmentation, the saturation of the FISH channel was increased to visualize cytoplasm contours. Overlapping cells were not quantified.”*

Reviewer Fig. 2. Maximum intensity projection of TERT smRNA FISH in iPS cells. Manual segmentation by thresholding the FISH channel was used to visualize cell contours.

6. Extended data Fig. 4a –

o It’s misleading to show these two genes side by side, since TERT is mostly long intronic sequence, not well conserved as expected, while TUG1 has relatively short introns. Maybe show just CDS of TERT, and then the relevant introns separately.

We agree with the reviewer and have added separate alignment of TERT CDS and introns to Figure 4a (previously Extended data Fig. 4a).

o This should be shown aligned to an RNA-seq track, to make it possible to assess the conservation of the retained introns.

This is a bit difficult to be done because gDNA sequences are stretched upon alignment (i.e., include large gaps), which makes it difficult to add the corresponding RNA-seq track.

o I'm not sure what the color gradient on the right means, if conservation here is per nucleotide it is not a continuous rather a binary variable. Can be removed or better explained.

We agree with the reviewer and have changed the color gradient.

7. Fig. 3C (right) should be presented same as Fig. 2C (right), this is more clear and informative. In addition, since both Fig. 2 and Fig. 3 are very similar and convey the same message, one of them can be moved to supplement.

This type or representation was applicable for TUG1 (Fig. 2C) due to larger dynamic range of nuclear enrichment of TUG1 between cell lines (35% to 70%) and more analyzed cell lines. However, nuclear enrichment of TERT is very high with little difference between cell lines (82%, 85%, 89%, 95%), followed by almost equally high retention of intron 11 (77%, 76%, 83%, 69%) (please see the reply to minor comment 2).

8. Extended data Fig. 5 - Splitting this figure to panels and adding references throughout the text will make the corresponding paragraph easier to follow

We split the figure to panel a and b, and referenced the text correspondingly (now Extended data Fig. 4).

9. Fig. 4 – This analysis does not provide substantial new insights, as splicing differences between coding genes and lncRNAs are known. It can be moved to the supplemental material.

The main message of the figure was not the difference in splicing between protein coding and lncRNA genes, but to show that retention of TERT introns is actually quite high among all protein coding genes (i.e., not typical), while TUG1 is rather typical for a lncRNA. Importantly, we aimed to convey that the extent of intron retention for both TERT and TUG1 in the corresponding gene class is conserved between human and mouse. We added a better suited description of these results to convey this point.

10. Line 288 – it seems that a few sentences were dropped here, since cell cycle was not mentioned up to this point? How was the cell cycle phase determined? In how many cells? Etc.

We agree with the reviewer that the transition can be explained better. To achieve this, we added an additional explanation at the beginning of the section to ease the transition to the analysis of TERT splicing in mitosis.

Lines 331-336: *“During our smRNA FISH analyses of intron 11 retention, we observed that TERT pre-mRNA was spliced during cell division (after late prophase). Specifically, we used DNA staining with Hoechst to distinguish cells in interphase and mitosis (either prophase, telophase, anaphase or prophase). We quantified unspliced TERT (co-localized intron 11 and exon signal), spliced TERT (exon signal only) and solo intron 11 in each stage after late prophase. In contrast to interphase, during mitosis...”*

11. In Fig. 7E, it seems clear that some alternative splicing is happening in both TUG1 introns. For intron 1, the TMO treatment clearly affects one of these alternative isoforms. In current RefSeq annotations these events are possibly represented by multiple isoforms. It is interesting to test using available RNA-seq data and/or RT-PCR the identity of these alternative isoforms, their subcellular localization, and the effect of TMOs on their efficiency.

The reviewer raises a very good point! Indeed, the agarose gel shows a clear shift in one of those isoforms of intron 1-spanning RT-PCR upon the TMO treatment. To determine the identity of those isoforms, we extracted the DNA from gel bands and performed Sanger sequencing. The results reveal an isoform with

a mini exon within intron 1 (isoform 2, labeled with * 2 in Figure below). TMO blocking the donor site reduces the abundance of isoform 2 towards isoform 3, which contains a small intron (labeled with * 3). These data have now been added in Extended data Fig. 9a.

We share the reviewer's interest in further discerning their subcellular localization. Ideally, we would apply the same approach to analyze the localization of isoform 3 as we did for other intron-retaining transcripts. Unfortunately, smRNA FISH approach cannot distinguish isoform 1 and 3. Synthesizing probes specific to isoform 3 is not feasible due to its short length (~260bp), which allows placing only around 10 probes (of recommended 48) when removing all filtering options (greatly increasing the probability of non-specific binding).

Nevertheless, the intron-spanning RT-PCR shows that the TMO approach mainly impacts the shift from isoform 1 to 4, thus we are confident that our work demonstrates that inclusion of full intron 1 influences the subcellular localization of TUG1.

Intron 2 contains at least 2 spliced isoforms whose donor and acceptor sites are in close proximity (depicted as band 3.1 and 3.2 in the initial Figure 7e, now placed in Extended data Fig. 9a). These small differences in splicing are not sufficient to differentiate their subcellular localization by smRNA FISH. The reviewer's proposal to get a deeper insight into the subcellular localization of all the isoforms is certainly interesting. The current study is mainly focused on discerning the localization and effect on localization of intron 1 and intron 2 retaining isoforms, therefore this is beyond the scope of this study. We feel that the proposed analysis would be more suitable as a part of future work dedicated to TUG1 only.

Extended data Fig. 9a. Additional isoforms identified with intron-spanning RT PCR. Arrow depicts main isoforms switching upon the TMO treatment. Star depicts less abundant isoforms changing upon the TMO treatment.

12. Line 341 – “Fig. 7F” (typo).

We thank the reviewer for pointing this out and we corrected the typo.

13. In Fig 7C you use “Cell growth assays”, while in Fig. 8B you use “Cell line viability assay” – are these different assays?

These are the same assays with Alamar blue, which primarily reflect cell viability. We changed the text to consistently refer to cell viability throughout the manuscript and figures.

Reviewer #2 (Remarks to the Author):

Dumbovic et al. study the nuclear retention of TERT mRNA and TUG1 lncRNA and show that intron retention is involved. Conservation of this mechanism between humans and mice is shown. They also develop a method to perturb intron excision that could have therapeutic use. I would recommend dividing this into two papers as the two RNAs studied don't seem to have a lot in common. If the authors don't want to do this, they need to at least expand discussion of TUG1 RNA to make it interesting. Further, they could extend this analysis by investigating the subnuclear location of the retained RNAs.

We are grateful to the reviewer for the positive assessment of our manuscript and their suggestions.

2.1. Nuclear retention of TERT mRNA by specific intron retention and its further splicing and export during mitosis is a very exciting and important finding. This could explain its activation in cancer cells. Further analysis of its subnuclear localization would be very interesting-is it in speckles? Chromatin?. It is obviously an important translated mRNA so nuclear retention must be a regulatory mechanism.

To address the reviewer's question we fractionated the HEK293T cells to nuclear insoluble (chromatin), nuclear soluble and cytoplasmic fractions and analyzed the distribution of intron-retained and spliced TUG1 and TERT. The results show that both intron-retained TUG1 and TERT have a tendency to be slightly more enriched in the chromatin fraction. However, they are found at higher abundance in the nuclear soluble fraction than a purely chromatin bound RNA such as U1 or Xist (Reviewer Fig. 3).

Furthermore, we looked into the publicly available data, poly(A)⁺ RNA sequencing of HEK293T nuclei fractionated into chromatin and nuclear soluble fractions. We observe the same tendency that nuclear TUG1 and TERT are more enriched in the chromatin fraction.

Reviewer Fig. 3. Nuclear TUG1 and TERT localize in nuclear soluble and chromatin fraction. **a**, Relative enrichment (in %) of TUG1 and TERT exon and intron-retaining transcripts in nuclear insoluble, nuclear soluble and cytoplasmic RNA fractions from HEK293T cells analyzed by RT-qPCR on oligo(dT) primed cDNA. Controls: GAPDH for cytoplasmic RNA, U1 for nuclear insoluble RNA. $N = 2$ or 3 replicas. **b**, Relative abundance of TERT and TUG1 in nuclear soluble and chromatin (nuclear insoluble) fractions of HEK293T nuclei assessed from poly(A)⁺ RNA-seq of the corresponding fractions. $N = 3$ replicas.

R2.2. TUG1 lncRNA is not described in much detail as to its normal function and localization. It is said to have a role in many cell processes and to be oncogenic. It doesn't seem unusual for an unspliced lncRNA to be nuclear and perhaps spliced and exported for degradation. Where in the nucleus is it located? Is it chromatin-associated? It's not clear what it has in common with TERT except both are up-regulated in cancer cells and have retained introns.

We understand the reviewer's concern and we extended the introduction to further explain the interest in analyzing both of these RNAs and to assess their nuclear localization in more detail. The mechanism and potential regulatory implications of retaining introns has not yet been shown for TUG1 or TERT. Furthermore, these RNAs provide an opportunity to test if unspliced introns are indeed sufficient to cause

nuclear retention, instead of simply being associated with nuclear retention, comparing a pre-mRNA and a lncRNA side-by-side.

Recent evidence showed that lowering the splicing efficiency of conserved lncRNAs between human and mouse can give rise to novel functions of RNA in the nucleus (Guo C. et al., Nature 2020). Hence, splicing efficiency of lncRNAs can have important notions for their functions. TUG1 RNA has been extensively studied up to date in the terms of its role in normal cellular physiology and cancer. However, up to date, there is no insight in the mechanism by which this lncRNA maintains dual localization in the nucleus and the cytoplasm. Although lncRNAs do have lower splicing efficiency compared to protein-coding genes and this is not unusual, nuclear localization of lncRNAs (for instance, MALAT1, NEAT1, Xist) is not necessarily accompanied by retention of introns, but other regulatory mechanisms have been suggested (for instance, Shukla C. et al., EMBO J. 2018, Lubelsky Y. & Ulitsky I., Nature 2018). Furthermore, as studies have showed, lower splicing efficiency does not necessarily mean stable retention of introns. Thus, our study provides a deeper insight into the mechanism by which the localization of TUG1 lncRNA is regulated and into the sequence variability of nuclear and cytoplasmic TUG1, which will have important implications in mechanistic studies involving TUG1.

R2.3. Line 99 (and 215) say that TERT intron 11 and 14 PIR are 30 and 31%, which made me wonder how that caused most RNA to be nuclear. Are intron 11 and 14 both retained on the same RNA or different RNAs? However, Line 122 says TERT intron 11 has a nuclear PIR of 90%. Please explain

Indeed, we have to note the difference in the method by which these values were obtained. PIR values of 30 and 31% were obtained using Vast-tools, which relies on analysis of poly(A) RNA-seq from bulk cells, compromising different cellular states. For instance, cells in mitosis have mainly spliced TERT, which will lower the total PIR value output from vast-tools. On the other hand, with single molecule RNA FISH, we directly measure the percent of intron retention in the nucleus (nuclear PIR) and in total RNA in a cell (total PIR) in interphase and mitosis. Nuclear PIR of 90% and predominantly nuclear localization were evaluated by single molecule RNA FISH in interphase cells, which in this sense has higher spatial resolution. To refer to this important concern, we added an explanation, lines 141-145:

“Of note, with smRNA FISH we can detect the absolute number of intron-retained and spliced transcripts per cell/compartiment in interphase and mitosis. Vast-tools analysis was done on bulk RNA from poly(A)-selected samples, therefore likely under-representing the pool of nuclear unspliced or partially spliced transcripts that we detect by smRNA FISH.”

R2.4. It appears that TERT intron 2 is also retained in some cell lines but is downplayed here. It seems possible that skipping exon 2 (and adjacent introns) would minimize its retention. Line 107: It seems like a poor control for a non-retained intron. What was the PIR for GAPDH intron 2?

We agree with the reviewer that intron 2 is retained in some cell lines. In particular, we found it retained in HCT116 cell line. Our intention was not to disregard this occurrence. On the contrary, we emphasized these results after analyzing RNA-seq and smRNA FISH of HCT116 cells:

Lines 173-174: *“RNA-seq analysis showed high read coverage in introns 11 and 14 (and in HCT116 cells, intron 2)...”*

Lines 187-195: *“While intron 2 was not significantly retained in other cell lines examined, HCT116 retained intron 2 in nuclear TERT (nuclear PIR=78%, $R^2=0.91$ with quantity of nuclear TERT, $P<2.2 \times 10^{-16}$, Pearson correlation), alongside intron 11 (nuclear PIR=69%, $R^2=0.86$ with quantity of nuclear TERT, $P<2.2 \times 10^{-16}$, Pearson correlation), while intron 14 was retained less efficiently (nuclear PIR=40%, $R^2=0.50$ with quantity of nuclear TERT, $P=6.5 \times 10^{-9}$, Pearson correlation). This atypical retention of intron 2 can also be observed in the corresponding RNA-seq for HCT116 (Fig. 3a).”*

Response to Reviewers Nature communications NCOMMS-20-27782A

Reviewer Fig. 4. GAPDH and TERT TS histogram showing the number of exon/intron 2 overlap in iPS cells.

We used TERT intron 2 as a control for the smRNA FISH setup in iPS cells, where it is not retained. This can be seen from the histogram (Reviewer Fig. 4) and as shown in the Fig. 1f (PIR=18%).

Typically, there are 1-2 dots per nucleus, which is what would be expected for a transcription site. GAPDH transcription sites (GAPDH exon/intron 2 overlap) are shown in Extended data Fig. 1c and we included the histogram in Reviewer Fig. 4 for convenience. We did not calculate the PIR for GAPDH; due to the very high quantity of the cytoplasmic GAPDH (hundreds of RNA molecules), while TERT has ~10 molecules per cell, these numbers would not be comparable. Thus, we refer to the histogram for showing the TS.

R2.5. Page 5: what does it mean that TUG1 intron 2 is absent from the VastDB human database? Is it not retained?

Please refer to our response to Reviewer #1's minor point 4.

R2.6. How much mouse TERT mRNA is retained in the nucleus? I couldn't find this info.

To address the reviewer's question whether mouse *Tert* mRNA is nuclear localized, we analyzed the enrichment of *Tert* using poly(A)⁺ RNA-seq from nuclear soluble, chromatin and cytoplasmic RNA fractions from mouse embryonic stem cells. The analysis shows that in mES cells, *Tert* is in fact predominantly nuclear. We added this data in Figure 4d.

Figure 4d. Relative subcellular localization of *Tert* and *Tug1* in poly(A)⁺ RNA-seq from chromatin, cytoplasm and nucleoplasm of mES cells. Cytoplasm-enriched *Gapdh* and *Actb* and chromatin-enriched *Firre* are plotted for comparison.

R2.7. It is interesting that many other mRNAs are retained in the nucleus. Are they cell cycle regulated also?

This was also one of our interests when analyzing the data. Unfortunately, we did not find any cell cycle regulated genes or significant GO terms that would bring together this high retention of introns in mRNAs in iPS cells.

R2.8. Extended data Table 3b: Some bedgraphs appear to be missing.

We appreciate the reviewer's attention to detail. We would like to add this, but we are not sure which bedgraphs are missing?

R2.9. Line 104: Tiling not tilling

We thank the reviewer for pointing this out and we have corrected the typo.

Reviewer #3 (Remarks to the Author):

In this manuscript, Dumbovic et al. compare the nuclear/cytoplasmic distribution of fully spliced RNAs to their intron-retained counterparts. They use smRNA FISH as their primary technique to investigate localization of TERT mRNA and TUG1 ncRNA. Overall, the data are clear, believable, and well presented in a nicely focused paper. The writing is generally good, and I do not have any major criticisms of the work that is presented. The work is descriptive and none of the observations are particularly surprising. In fact, unspliced introns leading to nuclear retention has been explored since the late 80s with the pioneering work from Sharp and Rosbash labs and recent global analyses suggest that intron retention is strongly associated with nuclear retention in mammals. Therefore, one may quibble about whether this work warrants publication in non-specialist journal like Nature Communications. Nonetheless, I would say the strength of the scientific approach and the clarity of the data support its publication there.

We thank the reviewer for the positive assessment of our manuscript and their suggestions.

I do have a few minor, but important, suggestions for improvement:

R3.1. In Fig 1 and throughout. Please use the figure legends and more intuitive axis labels to clarify what the graphs are. Just a few examples: What is "signal count" (same as "spots" used later)? Does "FISH signal" mean intensity or number of spots and is it the same as "signal count"? I have no idea what is being plotted in the middle graphs on Fig 1e and 1f. The grey and purple make for nice depiction of data and I pretty sure that I understand the point, but I have no idea what is being plotted. The X-axis is labeled "nucleus"; what does that mean? Does an N=50 mean number of cells or spots counted? If the latter, it seems like a small number of cells to count for this kind of analysis without demonstrating that these are representative cells. How many independent replicates of the experiment was performed?

We agree with the reviewer and accordingly changed axis labels to clearly annotate the labels. We changed all the axis labels for RNA FISH signal quantification to "FISH spots/compartiment" and clarified the figure legend. N depicts the number of cells quantified, in this example $n = 50$ means a total of 50 cells were quantified. Each FISH quantification was performed on independently acquired fields from at least 2 independent RNA FISH stainings.

R3.2. In the RNA-seq data and RT-qPCR data, please state in the Results or Figure legends whether or not the RNAs were primed by dT or random priming. This makes a big difference in the interpretation of whether these are fully processed RNAs with retained introns or whether they are potentially RNAs being transcribed and not yet processed.

The RT qPCR data shown in the results was synthesized with random primers. We agree with the reviewer that answering whether intron-retained transcripts are processed RNAs with a poly(A) tail is an important aspect to address.

To this end we compared the enrichment of intron-retained TUG1 and TERT transcripts on oligo(dT) and random primed cDNA during the iPS ActD time course. We used GAPDH as a control transcript with a poly(A) tail, and GAPDH intron 2 and 45s rRNA as controls for nascent and non-poly(A) containing transcripts. The analysis shows that both TUG1 and TERT intron-retaining transcripts have a poly(A) tail, indicating these are 3'-end processed transcripts. This data has been added to Figure 5b. Corresponding text has been added to Results, lines 299-303; and a new Methods section has been added.

Figure 5b. Relative abundance and stability of spliced and intron-retained transcripts in cDNA synthesized with random primers or oligo(dT) during the 4.5 h ActD treatment of iPS cells. GE: GAPDH exon, GI: GAPDH intron; E: exon; I: intron.

RNA-seq analyses were performed on poly(A) enriched libraries. The corresponding information has been added to the Results section, Figure legends and Methods section.

R3.3. One should be cautious regarding the ActD experiment interpretation. Nuclear RNAs with retained introns are hyperadenylated and may be artificially stabilized upon transcription inhibition (e.g. Bresson et al. 2015).

We added a short paragraph with this clarification in the results section, lines 325-327:

“We note that nuclear RNAs with retained introns were shown to be hyperadenylated upon transcription inhibition, which may contribute to their stabilization⁵⁹.”

R3.4. Line 288 is confusing—what is being referred to by “the above smRNA FISH analyses revealed that TERT pre-mRNA was spliced during cell division”?

We have extended the introduction to this section to ease the transition to the analysis during mitosis. Please see response to Reviewer #2, point 10.

R3.5. The interpretations of the TERT/mitosis data are too strong. There is a robust correlation and their data are believable, but without orthogonal approaches, it is too strong to say that “nuclear TERT transcripts uniformly and robustly retain two specific introns whose splicing occurs during mitosis”. Their data form a solid basis for this hypothesis, but it’s far from fully proven. It’s fine to include the data, but softening the interpretations is necessary.

We have softened the interpretation, line 23:

“Our data suggest that the splicing of TERT retained introns occurs during mitosis.”

R3.6. Line 341: 7f not 6f

We have corrected the typo.

R3.7. The statement on detained introns (line 432) implies that detained introns are defined as those that serve as reservoirs for posttranscriptional splicing. While some certainly do (e.g. Mauger et al. 2016), the definition of “detained” by Boutz et al. (2015), is simply that these are transcripts that have retained introns that are not exported to the cytoplasm. In principle, such RNAs could be reservoirs, nonfunctional partially spliced byproducts, or serve as nuclear noncoding RNAs.

The reviewer makes an excellent point. We discuss the potential of intron retention to diversify lncRNA functionality elsewhere in the discussion. To be specific that this was shown for some protein coding transcripts, we clarify the sentence highlighting that this was shown for some transcripts, lines 488-490:

“While such RNAs could be nonfunctional, partially spliced byproducts, or serve as nuclear non-coding RNAs, it has been shown that some of the nuclear, intron-retained RNAs are poised or ‘detained’ for a signal for post-transcriptional splicing, hence serving as a reservoir of RNAs readily available depending on cellular activity^{12,13}.”

REVIEWERS' COMMENTS

Reviewer #1 (Remarks to the Author):

The authors have address all the comments from the previous round of review in a satisfactory manner, which has improved the manuscript, and so I can now recommend publication in Nature Communications.

Reviewer #2 (Remarks to the Author):

The revised manuscript is now acceptable for publication.

Karen Beemon

Reviewer #3 (Remarks to the Author):

The authors have appropriately responded to all of my original critiques.